# Effortless Cross-Platform Video Codec: A Codebook-Based Method

## Abstract

Under certain circumstances, advanced neural video codecs can surpass the most complex traditional codecs in their rate-distortion (RD) performance. One of the main reasons for the high performance of existing neural video codecs is the use of the entropy model, which can provide more accurate probability distribution estimations for compressing the latents. This also implies the rigorous requirement that entropy models running on different platforms should use consistent distribution estimations. However, in cross-platform scenarios, entropy models running on different platforms usually yield inconsistent probability distribution estimations due to floating point computation errors that are platform-dependent, which can cause the decoding side to fail in correctly decoding the compressed bitstream sent by the encoding side. In this paper, we propose a cross-platform video compression framework based on codebooks, which avoids autoregressive entropy modeling and achieves video compression by transmitting the index sequence of the codebooks. Moreover, instead of using optical flow for context alignment, we propose to use the conditional cross-attention module to obtain the context between frames. Due to the absence of autoregressive modeling and optical flow alignment, we can design an extremely minimalist framework that can greatly benefit computational efficiency. Importantly, our framework no longer contains any distribution estimation modules for entropy modeling, and thus computations across platforms are not necessarily consistent. Experimental results show that our method can outperform the traditional H.265 (medium) even without any entropy constraints, while achieving the cross-platform property intrinsically.

## 1 Introduction

In recent years, neural network-based video codecs have attracted much attention in academia and industry. The rate-distortion (RD) performance of the latest neural video codecs (NVCs) has exceeded that of state-of-the-art traditional video codecs (e.g., H.266/VTM) to some extent (Bross et al., 2021; Wang et al., 2023; Li et al., 2023; 2022). An important reason why these NVCs are able to achieve high performance is the use of the entropy model, which improves the metrics by modeling the temporal redundancy information in the previous frame and the spatial correlation within the current frame, thus reducing the redundancy information in the compression process (Lu et al., 2019; Agustsson et al., 2020; Li et al., 2021a; 2022; 2023; Lin et al., 2020; Rippel et al., 2021; Wang et al., 2023; Zou et al., 2022; Xiang et al., 2022; Yang et al., 2020; Dosovitskiy et al., 2015; Ranjan & Black, 2017; Sun et al., 2018; Hui et al., 2018; Wang et al., 2023; Liu et al., 2022; Yang et al., 2023; Yang & Mandt, 2022; Kwan et al., 2023; Salman Ali et al., 2023).

However, designing a *cross-platform* NVC that can be applied in practice still faces a serious challenge. In cross-platform scenarios, most entropy model-based video codecs face non-deterministic computational problems, such as the incorrect reconstruction in Fig. 1. The non-deterministic computational problem is a common problem caused by floating point operations on different hardware or software platforms, because floating point errors on different platforms obey different distributions. As initially defined by Ballé et al. (2019), the non-determinism problem in cross-platform scenarios cannot be avoided when arithmetic coding is used for data compression. The principle is that different platforms introduce different systematic errors in the entropy modeling (Ballé et al., 2016; 2018; Li et al., 2022; Wang et al., 2023; Li et al., 2023). Specifically, as shown in Fig. 1, we separate the conventional neural compression pipeline into encoder and decoder sides. At the

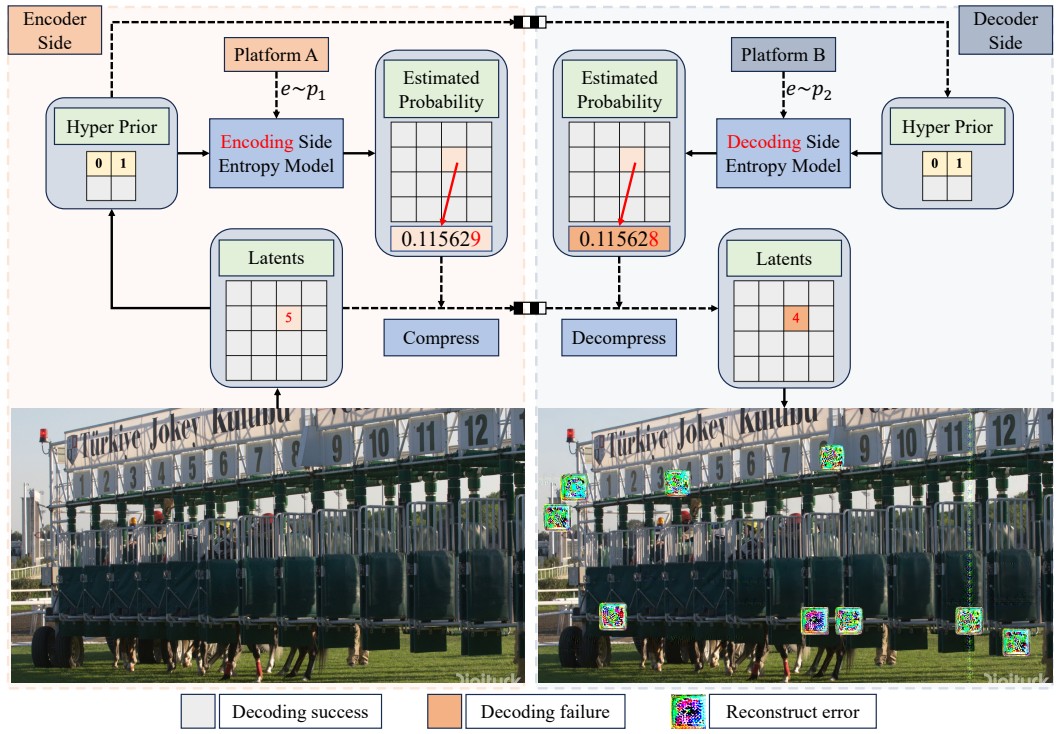

Figure 1: When the encoder and decoder run in cross-platform scenarios, the decoder will reconstruct an incorrect image on account of floating point math.

encoder side, the hyperprior is handled by the entropy model to obtain the estimated distribution of the latents, thus compressing the latents into the bitstream with a high compression ratio. At the encoding end of Platform A, the systematic error $e$ that follows the distribution $p_1$ is introduced by the entropy model, which will introduce another systematic error $e \sim p_2$ in the decoding end of Platform B. For the same hyperprior, computations at different ends yield slightly different estimates of the probability distribution, which results in the inability to recover the latent from the compressed bitstream completely and correctly using arithmetic decoding. Finally, it will lead to reconstruction errors, as shown in the bottom right image of Fig. 1.

Existing methods address non-deterministic problems mainly through quantization techniques, i.e., replacing uncertain floating point calculations with deterministic integer calculations (Ballé et al., 2019; Sun et al., 2021; Koyuncu et al., 2022; He et al., 2022). Nevertheless, all these methods require data calibration, which makes it complicated to deploy. Moreover, current neural network frameworks (e.g., Pytorch and TensorFlow) cannot support full-int computation, and thus custom implemented modules are necessary. There is also another solution to ensure data consistency between different platforms by transmitting calibration information, which yields a significant performance degradation if the systematic errors across different platforms are very large (Tian et al., 2023).

In this paper, we propose a codebook-based video compression framework by encoding video as index sequences of codebooks. Since no distribution estimation is required, there is no cross-platform problem theoretically. When we receive the correct index sequences on any decoding side, the decoding process is simplified to a codebook-based reconstruction problem.

Specifically, to achieve more effective information compensation, we design different codebooks for intra-frame (i.e., keyframe) and inter-frame (i.e., predicted frame) compression. For keyframes, we rely only on spatial redundancy information for image compression, with the codebook focusing on image reconstruction. For predicted frames, we use the reconstructed latents of the previous frame as the reference. Then, the required bitstream is reduced by another codebook that emphasizes information compensation. In order to better compress the predicted frames, we use a context model based on cross-attention to achieve the integration of latents from multiple frames. The computa-

tional complexity of the attention layer is reduced with a window-based strategy. Furthermore, the entropy model-based video compression frameworks accomplish different compression ratios by trading off between rate and distortion. In contrast, we achieve control over different bitrates and distortions by modifying the size of the codebooks.

Our proposed framework has several advantages over existing neural video codecs.

• We have addressed the cross-platform problem completely, as probability distribution estimation is no longer present in our method.

• We propose a cross-attention-based context model for temporal redundancy and spatial redundancy fusing. Due to the absence of autoregressive modeling and optical flow alignment, our proposed framework is extremely minimalist, which can greatly benefit computational efficiency.

## 2 RELATED WORK

### 2.1 CROSS-PLATFORM PROBLEM IN VIDEO COMPRESSION

The problem of computational inconsistency of image compression models in cross-platform scenarios was first identified by Ballé et al. (2019). They analyzed the reasons why other insensitive methods cannot avoid this common problem, since most encoding and decoding algorithms use arithmetic coding for data compression (Duda, 2009; Witten et al., 1987; Howard & Vitter, 1994; Cui et al., 2021; Guo et al., 2021). Consequently, to avoid floating point math in cross-platform, they proposed an integer-arithmetic-only network designed for learning-based image compression. A more complex entropy model, which is based on a Gaussian Mixed entropy model (GMM) and context modeling, was quantified by Koyuncu et al. (2022). He et al. (2022) used post-training quantization (PTQ) to train an integer-arithmetic-only model, thus enabling a general quantization technique for image compression. Existing methods are very similar to general model quantification techniques, classified as post-training quantization (PTQ, (Nagel et al., 2019; 2020; Li et al., 2021b)) and quantization-aware training (QAT, (Jacob et al., 2018; Esser et al., 2019; Bhalgat et al., 2020; Krishnamoorthi, 2018; Sun et al., 2021)). Tian et al. (2023) proposed calibration information transmitting (CIT) strategy, which encodes the error-prone entropy parameter coordinates into the auxiliary bitstream, to achieve consistency between the encoder and decoder.

### 2.2 NEURAL VIDEO COMPRESSION

DVC initially replaces all the components in the traditional hybrid video codec with an end-to-end neural network (Lu et al., 2019). DVC-Pro employs a more efficient network for residual/motion compression and the corresponding refinement network (Lu et al., 2020). To better handle the case of content loss and dramatic motion, Agustsson et al. (2020) proposed a scale-space flow that extends the optical flow-based estimation to 3D transformation. Hu et al. (2020) used multi-resolution instead of single-resolution compression of motion vectors to optimize rate-distortion.

Derived from the residual coding, DCVC employs contextual coding to compensate for the shortcomings of residual coding (Li et al., 2021a). Mentzer et al. (2022) proposed transformer-based temporal models for explicit motion estimation, warping, and residual coding. AlphaVC introduces various techniques to improve the rate-distortion performance, such as conditional I-frame and pixel-to-feature motion prediction (Shi et al., 2022). Li et al. (2022) used multiple modules, such as learnable quantization and parallel entropy model, to greatly improve the performance surpassing the latest VTM codec. MobileCodec is the first-ever inter-frame neural video decoder to run in real-time on a commercial mobile phone, regardless of the cross-platform considerations (Le et al., 2022).

### 2.3 VECTOR QUANTIZATION

Most of the existing works of Vector Quantization (VQ) focus on image representation and generation. Van Den Oord et al. (2017) proposed the Vector Quantized Variational Autoencoder (VQVAE), a method for learning a discrete representation of an image. Esser et al. (2021) used an encoder-decoder structure to train a quantizer that embeds images into compact sequences using discrete tokens from the learned codebook.

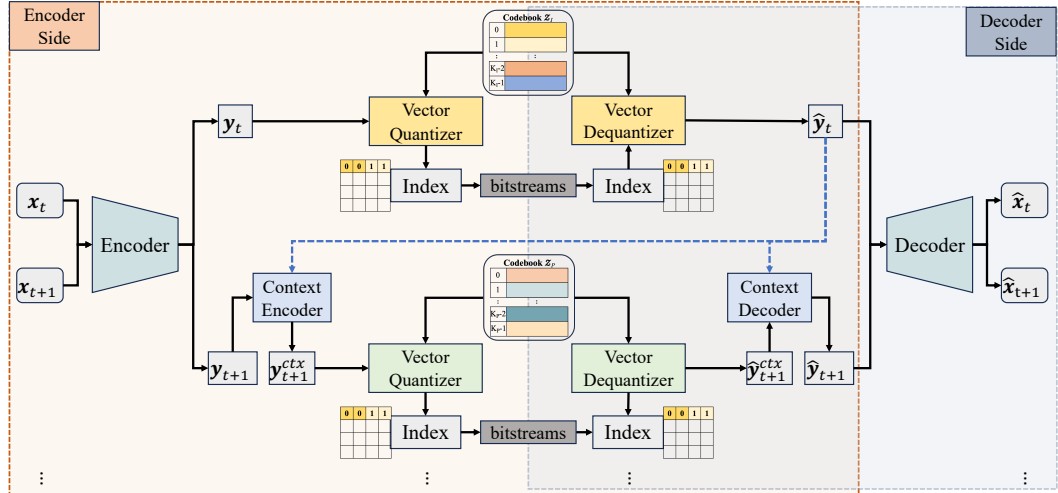

Figure 2: Our method is categorized into keyframes and predicted frames. We propose a cross-attention-based context model for temporal redundancy and spatial redundancy fusing. Due to the absence of autoregressive modeling and optical flow alignment, our proposed framework is minimalist. Keyframes and predicted frames depend on different codebooks for vector quantization.

Some research focuses on the compression of images using VQ. Duan et al. (2023) proposed a hierarchical quantized VAE for coarse-to-fine image compression that narrows the gap between image compression and generation. Kang et al. (2022) obtained the image residual through a three-way autoregressive model and used VQ-VAE as an entropy model for distribution estimation to achieve image compression. Without an entropy model, Zhu et al. (2022) proposed a multi-stage multi-codebook method for image compression.

## 3 PROPOSED METHOD

Our framework is designed accordingly for keyframes $x_t$ and predicted frames $x_{t+1}$, as shown in Fig. 2. Usually, we encode and decode the video in a group of pictures (GOP), the first frame within each GOP is the keyframe, and the other frames are predicted frames.

For keyframes, we use codebook-based methods for image compression (Zhu et al., 2022). While, for the more important predicted frames in video compression, we propose a cross-attention-based context model for temporal redundancy and spatial redundancy fusing, which no longer relies on optical flow alignment and autoregressive modeling. Window-based attention is presented to reduce the computation effort. Specifically, the context encoder integrates the reference latents $\hat{y}_t$ with the current latents $y_{t+1}$ to obtain the context information $y_{t+1}^{ctx}$, while the context decoder recovers the reconstructed latents $\hat{y}_{t+1}$ from the reference latents $\hat{y}_t$ and the quantized context $\hat{y}_{t+1}^{ctx}$.

It is notable that we use the same encoder and decoder in the keyframes and predicted frames to ensure the consistency of the reference latents. Benefiting from the cross-attention interaction, the reference latents and current latents are not necessarily well aligned, and thus registration based on optical flow could be omitted.

### 3.1 FRAMEWORK OVERVIEW

As shown in Fig. 2, our model contains three primary symmetric components, *encoder* and *decoder*, *context encoder* and *context decoder*, and *vector quantizer* and *vector dequantizer*.

**Image encoder and decoder.** For an input frame $x \in \mathbb{R}^{H \times W \times 3}$, we obtain the latents $y$ using $y = \mathrm{E}(x) \in \mathbb{R}^{h \times w \times n_c}$, where $(H, W)$ is the image resolution, and $(h, w, n_c)$ represents the dimensions of latents. Then, $y$ is quantized to $\hat{y}$ by vector quantization. For the quantized latents $\hat{y}$, we decode

the reconstructed image $\hat{x}$ by using $\hat{x} = \mathrm{D}(\hat{y}) \in \mathbb{R}^{H \times W \times 3}$. $\mathrm{E}(\cdot)$ is the image encoder and $\mathrm{D}(\cdot)$ is the image decoder for both keyframes and predicted frames.

**Vector quantization and dequantization.** We transform latents $y$ to the codebook index sequence $s \in \mathbb{R}^{h \times w}$, where $s^{ij} \in \{0, ..., K-1\}$ is the index from a learnable codebook $\mathcal{Z} \in \mathbb{R}^{K \times n_c}$ through quantization method, where $K$ is the codebook size, and $i, j$ are the element indices. It performs element-wise quantization $\mathrm{Q}(\cdot)$ of each spatial code $y^{ij} \in \mathbb{R}^{n_c}$ onto the nearest codebook index $k \in \{0, ..., K-1\}$ to obtain the index by

$$s^{ij} = \mathrm{Q}(y^{ij}, \mathcal{Z}) = \arg\min_k \left\| y^{ij} - \mathcal{Z}^k \right\|_2. \tag{1}$$

More precisely, the latents $y$ is represented by a sequence of indices $s$ from the codebook $\mathcal{Z}$ by $\mathrm{Q}(\cdot)$, which is denoted as *vector quantizer* in Fig. 2.

Symmetric with the *vector quantizer*, we recover the quantized latents $\hat{y}$ from the sequence of indices $s$ by dequantization method $\mathrm{deQ}(\cdot)$, replacing each index $s^{ij}$ with their corresponding codebook words $\mathcal{Z}^{s^{ij}}$, which is formulated as

$$\hat{y}^{ij} = \mathrm{deQ}(s^{ij}, \mathcal{Z}) = \mathcal{Z}^{s^{ij}} \in \mathbb{R}^{n_c}. \tag{2}$$

Specifically, the sequence of indices $s$ is reconstructed to the quantized latents $\hat{y}$ by $\mathrm{deQ}(\cdot)$, which is shown by *vector dequantizer* in Fig. 2.

**Context encoder and decoder.** Conditioned by the reference latents $\hat{y}_t$, the current latents $y_{t+1}$ is transformed into a context latents $y_{t+1}^{ctx}$ by the contextual encoder $\mathrm{ctxE}(\cdot)$ as

$$y_{t+1}^{ctx} = \mathrm{ctxE}(y_{t+1}, \hat{y}_t). \tag{3}$$

The current quantized latents $\hat{y}_{t+1}$ is parsed by contextual decoder $\mathrm{ctxD}(\cdot)$, conditioned on the same reference latents $\hat{y}_t$, which is given by

$$\hat{y}_{t+1} = \mathrm{ctxD}(\hat{y}_{t+1}^{ctx}, \hat{y}_t). \tag{4}$$

### 3.2 CROSS-ATTENTION-BASED CONTEXT MODEL

Different from existing methods that use entropy modeling and arithmetic coding to transmit latents, we compress video by transmitting the sequence of indices from the codebook. To make the latents represented by the codebook more informative and efficient, we employ a context model to integrate temporal redundancy (i.e., redundancy between current latents $y_{t+1}$ and reference latents $\hat{y}_t$) and spatial redundancy (i.e., redundancy between the neighborhoods of current latents $y_{t+1}$). The integrated latents are identified as $y_{t+1}^{ctx}$.

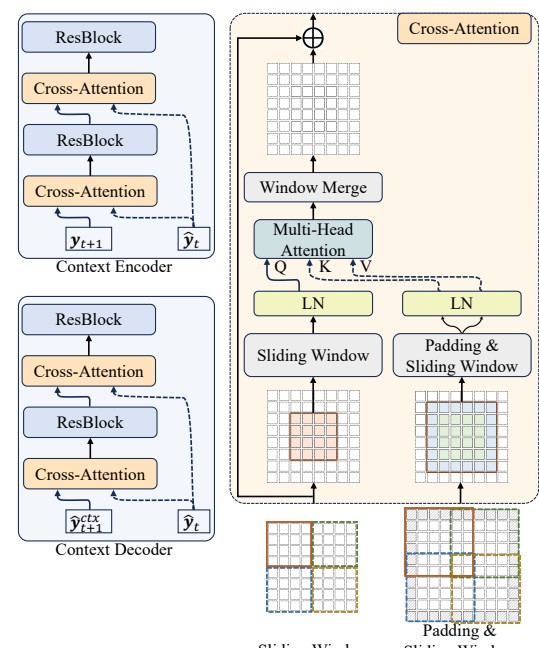

Figure 3: The context model architecture based on cross-attention layer. The structure of the cross-attention layer is described in detail, as well as the *sliding window* and *padding & sliding window* strategies that need to be used in it.

Specifically, as shown in Fig. 3, our context encoder is composed of the cross-attention layer and the resblock layer. Through the cross-attention layer, we encode the temporal redundancy contained in the reference latents $\hat{y}_t$, and then encode the spatial redundancy of the current latents $y_{t+1}$ through the local convolution of the resblock layer. Moreover, this process is repeated twice to achieve more sufficient context fusion.

Similar to the context encoder, the context decoder follows the same structure to combine the quantized context latents $\hat{y}_{t+1}^{ctx}$ and reference latents $\hat{y}_t$ to obtain the quantized latents $\hat{y}_{t+1}$ of the current frame.

**Cross-attention (CA).** Instead of using optical flow for context alignment, we propose to use the conditional cross-attention module to obtain the context information between frames. Specifically, the cross-attention takes the reference latents as the *key* and *value* while using the current latents as *query*, which no longer requires pixel-level context alignment. Due to the absence of autoregressive modeling and optical flow alignment, our framework is extremely minimalist which can greatly benefit computational efficiency.

**Window-based cross-attention (WCA).** The global cross-attention of the context model is computationally unaffordable for video compression with high resolution. We use window attention in the context model to alleviate this problem, as visualized in Fig. 3. Specifically, we first partition the current latents into non-overlapping small windows using the *sliding window* method, with a window size of $s_{sw}$. To encode the motion information between the reference frame and the current frame, we partition the reference latents into partially overlapping windows using a *padding & sliding window* method, with a window size of $s_{psw} = s_{sw} + 2 \times s_p$. Here, $s_p$ is the padding size.

### 3.3 MULTI-STAGE MULTI-CODEBOOK VECTOR QUANTIZATION

The core problem addressed by classical VQ-GAN and VQVAE is the discrete representation of an image, which is implemented in the form of the single-stage single-codebook VQ. To obtain high-quality reconstructed images without significantly increasing the size of the codebook and the number of indices that need to be transmitted, Zhu et al. (2022) proposed a multi-stage multi-codebook VQ method. Derived from Zhu et al. (2022), we also employ a multi-stage multi-codebook hierarchy for latent compression of video. Specifically, as shown in Fig. 4, the input latent $y$ is downsampled to obtain the latent $y_d$, which is then quantized by the first multi-codebook vector quantization layer (MCVQ) to obtain the quantized latent $\hat{y}_d$. Since the first MCVQ layer focuses on reconstructing the low-frequency signals, we use the next downsample and MCVQ layer to quantize the high-frequency residual $y'$ between $y_d$ and $\hat{y}_d$ to obtain $\hat{y}_d'$. We use the combination of downsampling and MCVQ three times to achieve progressive compensation from low frequency to high frequency. In addition, the low-frequency signals in an image require more bitstreams to encode than the high-frequency signals (Duan et al., 2023). Based on this principle, we use downsampling between different stages to achieve the spatial dimensionality decrease from low to high frequencies to reduce the number of indices, and gradually reduce the size of the codebook at each stage to minimize the bitlength of a single index.

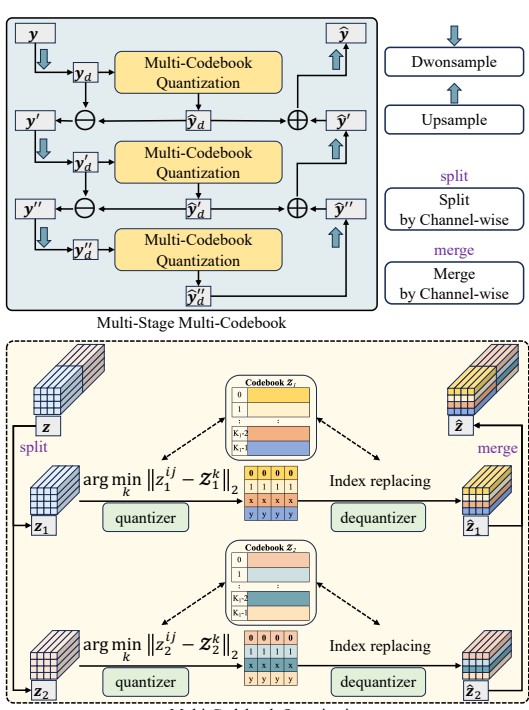

Figure 4: The multi-stage multi-codebook VQ architecture improves the reconstruction capability through successive compensation of codebooks at each stage. Multiple codebooks enhance the VQ representation at each stage.

While decoding, we reconstruct the quantized latents $\hat{y}$ from high-frequency latents $\hat{y}_d''$ to low-frequency latents $\hat{y}_d$ by progressively upsampling the quantized latents of the next level (e.g., $\hat{y}_d''$) and compensating the quantized latents of the previous level by addition (e.g., $\hat{y}'' + \hat{y}_d'$).

For the MCVQ layer, as shown at the bottom of Fig. 4, we first *split* the input latents $z$ in channel dimension to get $z_1$ and $z_2$. Then we quantize and inverse quantize $z_1$ and $z_2$ using codebooks $\mathcal{Z}_1$ and $\mathcal{Z}_2$ to obtain $\hat{z}_1$ and $\hat{z}_2$, respectively. Finally, $\hat{z}_1$ and $\hat{z}_2$ are combined in the channel dimension by *merge* to obtain the quantized latents $\hat{z}$.

### 3.4 Implementation Details

**Image encoder and decoder.** For both keyframes and predicted frames, we use the same image encoder $E(\cdot)$ and image decoder $D(\cdot)$ to transform between image $x \in \mathbb{R}^{H \times W \times 3}$ and latents $y \in \mathbb{R}^{h \times w \times n_c}$. In our experiments, we work with $h = \frac{H}{8}$, $w = \frac{W}{8}$, and $n_c = 128$.

**Window-based cross-attention.** Instead of using optical flow for context alignment, we propose to use the conditional cross-attention module to obtain the context information between frames. Finally, window-based cross-attention is used to reduce the computational effort, since attention is only considered within local windows. We set the sliding window size $s_{sw}$ of *query* to 4, and the padding size $s_p$ to 2. Then the sliding window size $s_{psw}$ of *key* and *value* will be $4 + 2 \times 2 = 8$.

**Codebook settings.** As shown in Fig. 4, the multi-stage multi-codebook vector quantization structure that we use in both keyframes and predicted frames employs a three-stage MCVQ and uses two codebooks during every stage of MCVQ.

Existing methods typically achieve different compression ratios by trading off loss weights between bitrate and distortion (Xiang et al., 2022; Li et al., 2022; 2021a; Lu et al., 2019). In contrast, we achieve different bitrates by modulating the codebook sizes used in the predicted frames. Specifically, we use a group of three parameters to specify different codebook sizes in multi-stage multi-codebook vector quantization. The codebook sizes for keyframes are specified as $\{8192, 2048, 512\}$, which are maintained constant at different compression ratios. While for predicted frames, we use three different groups of codebook sizes to achieve different video compression ratios as $\{8192, 2048, 512\}$, $\{64, 2048, 512\}$ and $\{8, 2048, 512\}$.

**Optimization Loss.** Distortion loss $\mathcal{L}$ is optimized throughout the training process, where $d(\cdot)$ represents the mean square error or MS-SSIM; $m$ is the GOP size used for training, where all frames are treated as predicted frames except the first frame, which is treated as a keyframe.

$$\mathcal{L} = \sum_{t=0}^{m} \underbrace{d(x_t - \hat{x}_t)}_{\text{distortion}} \tag{5}$$

## 4 Experiments

### 4.1 Experimental Setup

**Datasets.** We use Vimeo-90k dataset (Xue et al., 2019) for training, which contains 89800 video clips with the resolution of $448 \times 256$. The videos are randomly cropped into $256 \times 256$ patches. The training GOP size $m$ is set to 7 for each clip of Vimeo-90k containing 7 frames. We evaluate our model using UVG (Mercat et al., 2020), HEVC Class B, and MCL-JCV (Wang et al., 2016) datasets, which both have a resolution of $1920 \times 1080$ (1080P). The images are cropped to $1920 \times 1024$ by center cropping to ensure that the input image shape is divisible by 128.

**Baselines.** Neural network-based methods commonly rely on engineering optimizations to replace some modules with integer networks to achieve cross-platform capability. There are also methods that achieve cross-platform consistency by transferring calibration information through plug-ins. Conventional video codecs, including H.264, and H.265, are the only video codecs that can be practically used cross-platform. Different from other neural network-based methods, our algorithm is designed to avoid cross-platform problems. Therefore the conventional codecs H.264 and H.265 in FFmpeg are used as the anchor with *medium* preset.

**Metric.** We evaluate the performance of all models using common metrics including PSNR and MS-SSIM (Wang et al., 2004).

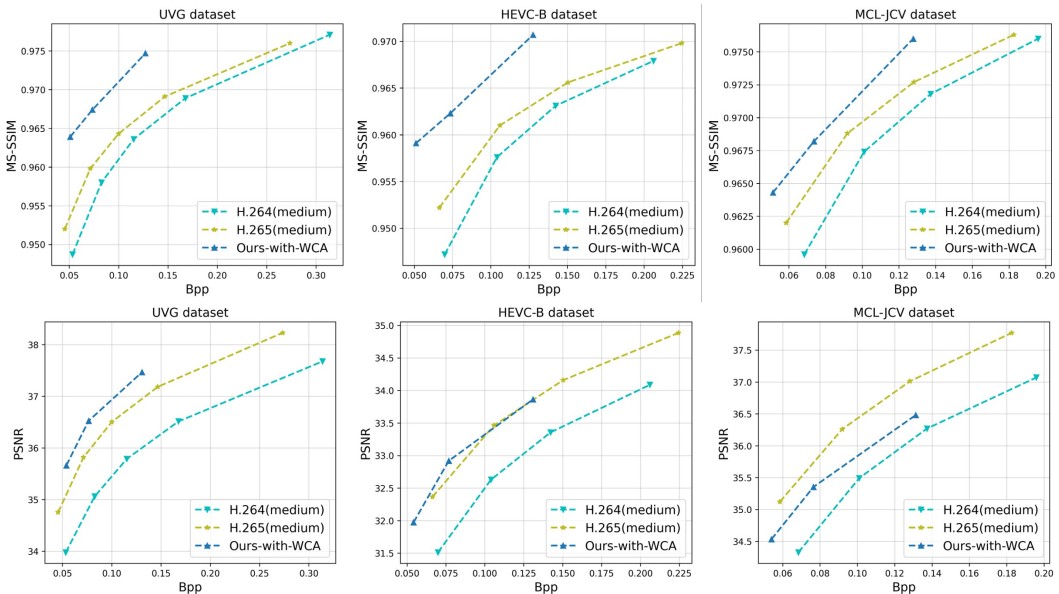

Figure 5: Rate-distortion performance on UVG, HEVC-B, and MCL-JCV datasets.

Table 1: BD-rate calculated by *SSIM* and *PSNR* on test datasets with the anchor H.265 (medium).

|  | SSIM-BPP | | | PSNR-BPP | | |
|---|---|---|---|---|---|---|
|  | H.264 | H.265 | Ours-w-WCA | H.264 | H.265 | Ours-w-WCA |
| UVG | +21.2 % | 0 % | -43.7 % | +63.7 % | 0 % | -23.7 % |
| HEVC-B | +18.4 % | 0 % | -38.2 % | +41.5 % | 0 % | -5.2 % |
| MCL-JCV | +19.9 % | 0 % | -19.1 % | +49.3 % | 0 % | +23.7 % |
| Average | +19.8 % | 0 % | -33.7 % | +51.5 % | 0 % | -1.7 % |

**Test conditions.** We test each video for 96 frames with GOP size 12 for H.264 and H.265, same in (Tian et al., 2023; Li et al., 2021a). For our model, we use GOP size 32 for testing. The experiments are conducted in cross-platform scenarios, where the videos are encoded with an NVIDIA Tesla V100 machine and decoded with an NVIDIA Tesla P40 machine. Since the existing neural video codecs cannot achieve cross-platform decoding directly, we only compare our method with the conventional codecs.

## 4.2 RESULTS

**Rate-distortion performance.** In Fig. 5, we plot the rate-distortion of our method and the baseline methods. Our method improves significantly over H.265 across different datasets in terms of SSIM. For PSNR, we outperform H.265 on UVG and HEVC-B datasets, but not on MCL-JCV. To analyze the reasons for the performance degradation on MCL-JCV, we show one of the videos with little variation and high redundancy *in the Appendix, Section B*. For videos with high redundancy, our method uses a fixed number of indices for video compression which results in performance degradation.

As shown in Table 1, we evaluate quantitative metric with BD-rate (Bjontegaard, 2001) computed from PSNR-BPP and SSIM-BPP, respectively. Our method achieves an average of 33.7% bitrate saving in terms of SSIM, while the improvement on PSNR is limited, with an average of 1.7% bitrate saving compared to H.265. Notably, our method achieves 23.7% bitrate saving on the UVG dataset with evident motion.

**Cross-platform results.** To verify the cross-platform capability, we encode the video at V100 using our model. On the decoding end, we compare the decoding results obtained from V100 and P40, respectively. We calculate the BD-rate for both decoders when decoding the bitstreams

Table 2: BD-rate calculated on different platforms by *SSIM* and *PSNR* on UVG dataset.

|  | SSIM-BPP | | PSNR-BPP | |
|---|---|---|---|---|
| Encoder | V100 | V100 | V100 | V100 |
| Decoder | V100 | P40 | V100 | P40 |
| UVG | 0 % | 0 % | 0 % | 0 % |

Table 3: Model complexity and BD-rate. All models are tested with 1080P video on V100.

|  | Params | MACs | Context time | Encoding time | Decoding time | SSIM BD-rate | PSNR BD-rate |
|---|---|---|---|---|---|---|---|
| Resblock-based | 53.438M | 2.221T | 3.2ms | 102.2ms | 219.5ms | 0% | 0% |
| CA-based-64 | 53.408M | 2.219T | 74.2ms | 242.9ms | 290.2ms | -40.8% | -48.0% |
| **WCA-based-4** | 53.408M | 2.233T | **13.0ms** | 122.4ms | **229.3ms** | -40.7 % | -49.0 % |
| light-decoder | 46.162M | 0.740T | 13.1ms | 122.2ms | **35.8ms** | -23.7 % | -37.4 % |

that are originally encoded by V100, which is shown in Table 2. These error-free results are a solid demonstration of the cross-platform capabilities of our method. *We also conduct experiments on the decoding side running at more platforms, which is shown in the Appendix, Section C.*

## 4.3 ABLATION STUDIES

**Window-based cross-attention.** Since the CA-based model will lead to out-of-memory, we use the *WCA-based* model with window size 64 as an alternative comparison option. We implement *Resblock-based*, *CA-based-64*, and *WCA-based-4* models, where *CA-based-64* is the WCA method with window size 64 and *WCA-based-4* is our proposed model. Experimental results in the last two columns of Table 3 show that our proposed *WCA-based* method can deliver bitrate saving of approximately 40% over the *Resblock-based* model in terms of SSIM. *More visualization results are provided in the Appendix, Section D.*

**Decoding efficiency.** Table 3 shows the complexity comparison of the models in the number of parameters, MACs (multiply-accumulate operations), context model time, encoding time, and decoding time (including context model time). All our models are tested on a server with an NVIDIA Tesla V100 GPU. Our proposed method saves 82.5% of context modeling time when similar performance is obtained using the *WCA-based* methods with different window sizes. It is worth noting that our *light-decoder* model, with fewer parameters, can decode 1080P video in real-time on V100 and still outperform H.265, as demonstrated in the last row of Table 3. *We provide further visualization in the Appendix, Section D and Section E.*

## 5 CONCLUSIONS

In this work, we present a video compression framework based on codebooks. Our method models temporal and spatial redundancy using a *WCA-based* context model, which avoids autoregressive modeling and optical flow alignment. The absence of the entropy model for probability distribution estimation makes our approach inherently effective across platforms, which is crucial for practical applications of neural video codecs. Our method outperforms H.265 in terms of SSIM on multiple datasets, and achieves better PSNR results than H.265 on the UVG and HEVC-B datasets. Our work provides valuable insights into the development of neural video codecs.

Our proposed method for compressing videos to a sequence of indices of the codebook has some limitations. Specifically, our method does not currently impose any constraints on the entropy, which can result in the model delivering constant bitstreams for different videos. As a result, when compressing videos with high redundancy, *as demonstrated in the Appendix, Section B*, our method may perform poorly. Consequently, we will consider designing entropy constraints for the indices of different videos to achieve content-based adaptive compression in future work.

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

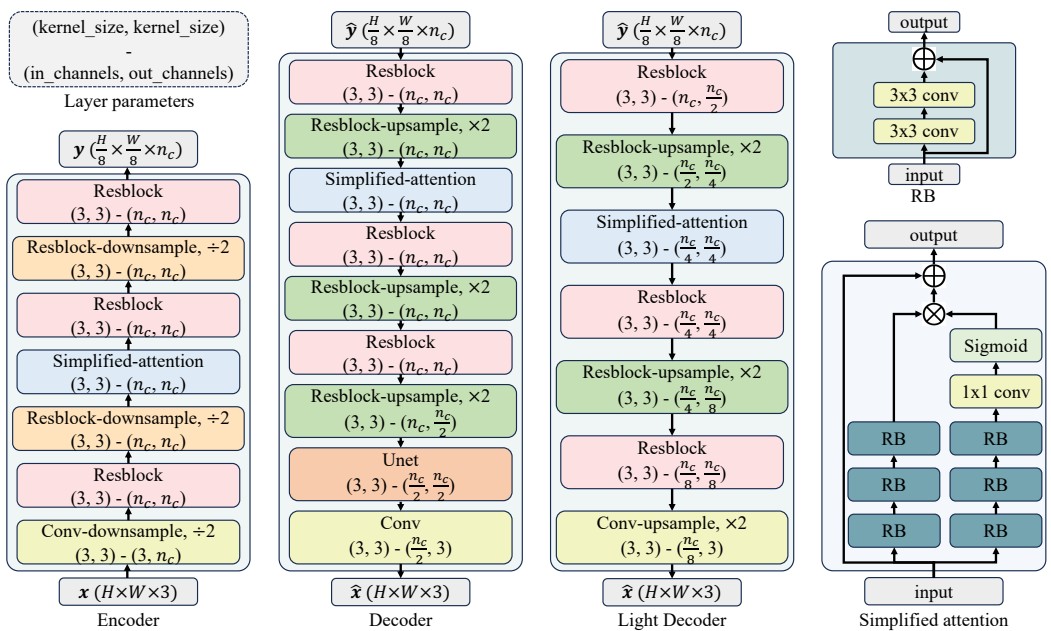

Figure 6: Image encoder/decoder architecture.

Table 4: Comparison of BPP for the same video in the same reconstructed PSNR condition.

| videoSRC30 | H.265 | Ours-with-WCA |
|---|---|---|
| PSNR | 37.589 | 37.483 |
| BPP | 0.033 | 0.054 |

## A  NETWORK

**Image encoder and decoder.** For both keyframes and predicted frames, we use the same image encoder $E(\cdot)$ and image decoder $D(\cdot)$ to transform between image $\boldsymbol{x} \in \mathbb{R}^{H \times W \times 3}$ and latents $\boldsymbol{y} \in \mathbb{R}^{h \times w \times n_c}$. The network architectures are shown in Fig. 6. In addition, we show the model architecture of *light decoder*.

**Context encoder and decoder.** Instead of using optical flow for context alignment, we propose to use the conditional cross-attention module to obtain the context information between frames. Specifically, for more efficient computation and sufficient context information fusing, we design the *WCA-based* context model with the architecture shown in Fig. 7. The architecture of the WCA layer is shown in Fig. 3. The number of *heads* is 8, and the *dimension of each head* is 16.

**Vector quantization and dequantization.** Derived from Zhu et al. (2022), we also employ a multi-stage multi-codebook hierarchy to vector quantization of the latent. More details about the architecture of multi-stage multi-codebook VQ can be found in (Zhu et al., 2022).

## B  VISUALIZATION OF SAMPLE VIDEO OF MCL-JCV

As shown in Figure. 8, we select a typical video sequence in the MCL-JCV dataset, in which the small target (i.e., the bird in the video) undergoes a slight motion. Our method uses a fixed number of indices for video compression, which inevitably wastes indices in videos with high redundancy, resulting in higher bitrates for the same reconstruction quality, as shown in Table 4.

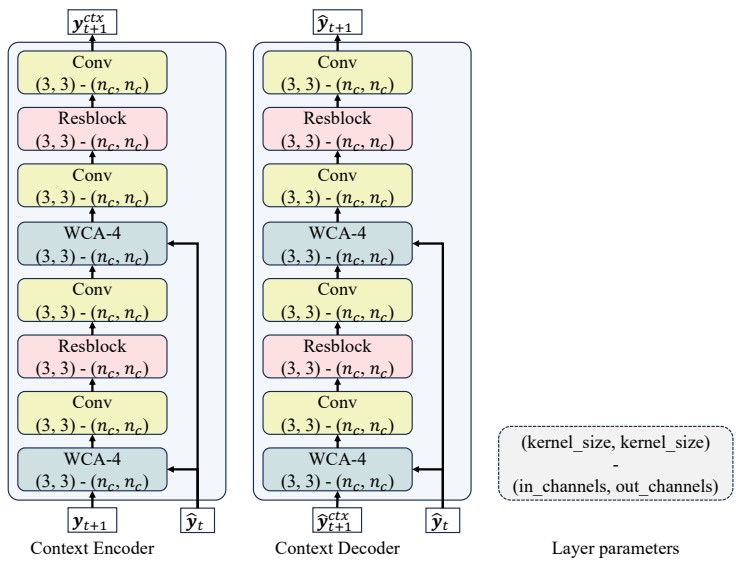

Figure 7: Context encoder/decoder architecture.

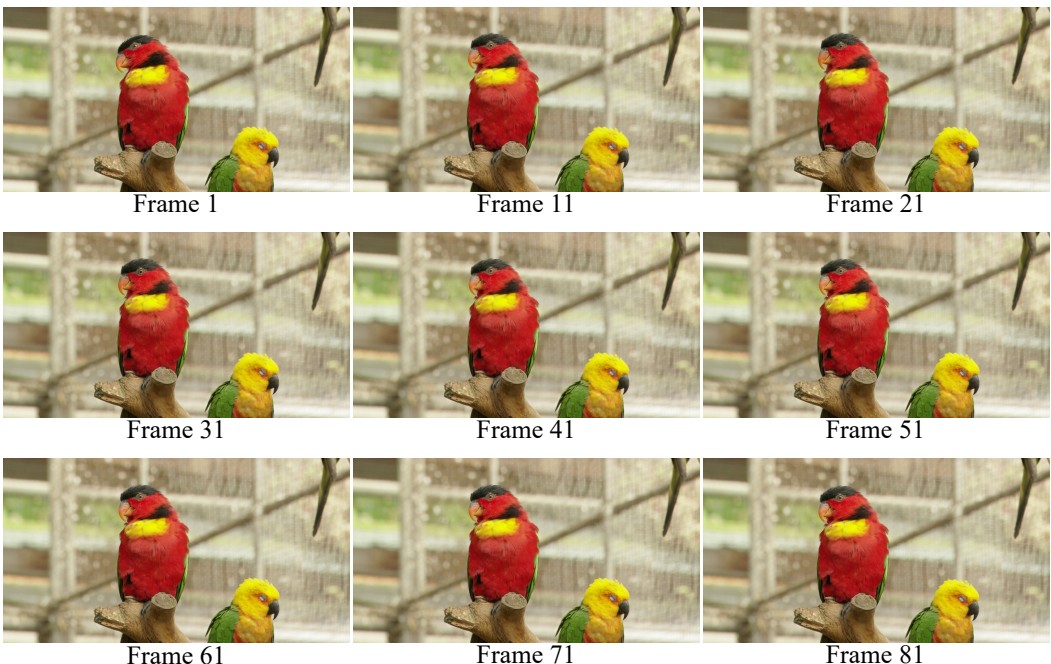

Figure 8: A sequence of frames sampled from video *videoSRC30* of the MCL-JCV dataset, which has high redundancy.

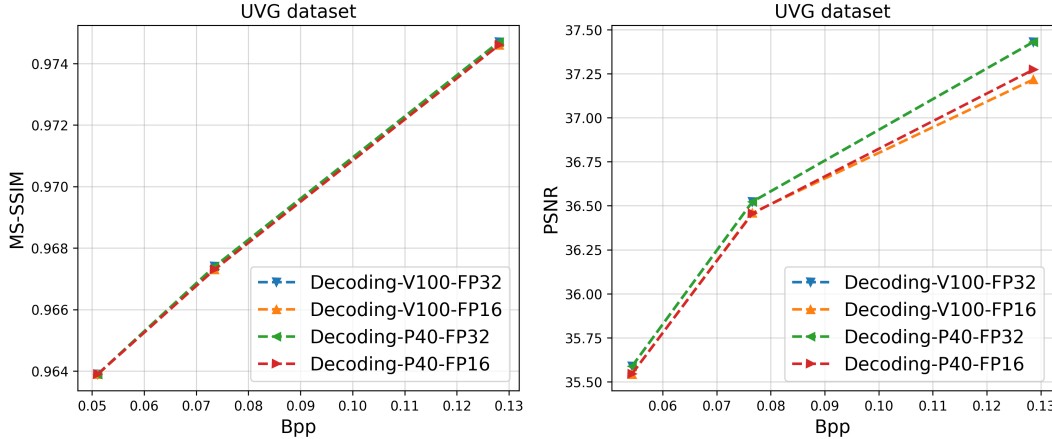

Figure 9: Rate-distortion of our method under different decoding platforms on the UVG dataset.

Table 5: BD-rate calculated on different platforms by *SSIM* and *PSNR* on UVG dataset.

| | SSIM-BPP | | | | PSNR-BPP | | | |
|---|---|---|---|---|---|---|---|---|
| Decoder precision | V100 FP32 | V100 FP16 | P40 FP32 | P40 FP16 | V100 FP32 | V100 FP16 | P40 FP32 | P40 FP16 |
| UVG | 0 % | +0.9 % | 0 % | +0.9 % | 0 % | +4.2 % | 0 % | +3.9 % |

## C   CROSS-PLATFORM ALATION

To verify the cross-platform capability, we encode the video at V100 using our model, which runs with FP32 precision (i.e., single-precision floating point format). On the decoding side, we chose V100 and P40 to decode using FP32 and FP16 (i.e., half-precision floating point format), respectively. In Fig. 9, we plot the rate-distortion of our method under different decoding platforms. For our cross-platform settings, the difference between FP32 and FP16 is very large, and our method is still able to reconstruct all frames correctly, which further demonstrates the cross-platform capability of our approach. Compared to the FP32 decoding results on V100, we calculate the BD-rate of the decoding results under other conditions, which is shown in Table 5.

It is worth noting that the decrease in BD-rate when decoding with FP16 precision, compared to decoding with FP32 precision, is attributed to our model being trained at FP32 precision. This difference in BD-rate is not a cross-platform discrepancy but rather a result of the disparity in computational accuracy between FP16 and FP32.

## D   WINDOW-BASED CROSS-ATTENTION ABLATION

In Fig. 10, we show the performance of our method when the context model is based on different approaches. It can be seen that our proposed *WCA-based* context model can bring significant improvement compared to the *Resblock-based* context model. It is worth noting that a *light-decoder* model with fewer parameters can decode 1080P video in real-time on V100 and still outperform H.265.

## E   RATE-DISTORTION CURVES FOR DIFFERENT GOP SIZES.

**Larger GOP.** Our approach uses a *WCA-based* context model for context fusion, which allows us to employ a larger GOP for compression. We evaluate the UVG dataset at different GOP sizes. The rate-distortion curves are shown in Fig. 11. It has been observed that the model's performance demonstrates a noteworthy enhancement as the GOP size is elevated from 7 to 32. However, there seems to be no substantial alteration in the model's performance as the GOP size is further increased

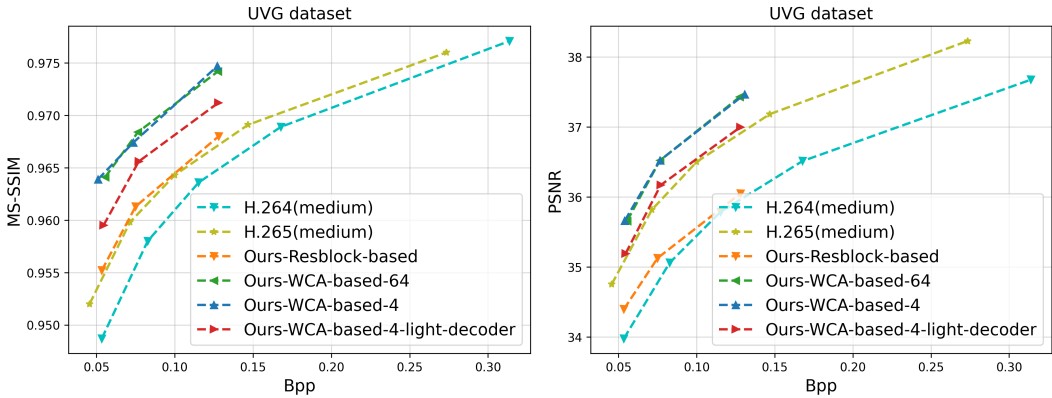

Figure 10: Rate-distortion of different context models on UVG dataset.

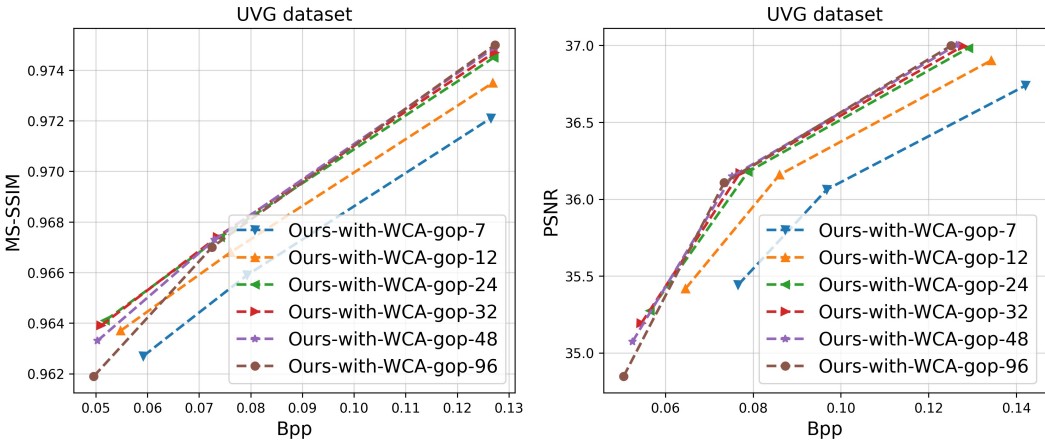

Figure 11: Rate-distortion for different GOP sizes on UVG dataset.

from 32 to 96. Table 6 displays the BD-rate for various GOP sizes, with a GOP size of 32 as the baseline.

## F  MORE PERFORMANCE COMPARISON.

To further demonstrate the effectiveness of our method, we conduct experiments on additional datasets, including HEVC Class C (480P), D (240P), and E (720P). Beyond PSNR and MS-SSIM, we introduced the perceptual evaluation (Andersson et al., 2020) metric LPIPS (Zhang et al., 2018) as a supplementary experiment. In addition, we compare the performance of different GOP sizes (e.g., 12 and 32), which are commonly used in present methods (Li et al., 2021a; Lu et al., 2019; Li et al., 2022; Tian et al., 2023; Li et al., 2023; Sheng et al., 2022). Furthermore, for H.264 and H.265, we analyze the impact brought by quality control parameters (e.g., *qp* and *crf*). The detailed FFMPEG settings are shown in the Appendix, Section G.

Table 6: BD-rate for different GOP sizes on UVG dataset.

| GOP size | 7 | 12 | 24 | 32 | 48 | 96 |
|---|---|---|---|---|---|---|
| SSIM-BPP | +21.9% | +9.0 % | +0.9 % | 0 % | -0.2 % | +0.5% |
| PSNR-BPP | +32.4% | +12.3 % | +1.9 % | 0 % | -1.3 % | -1.8 % |

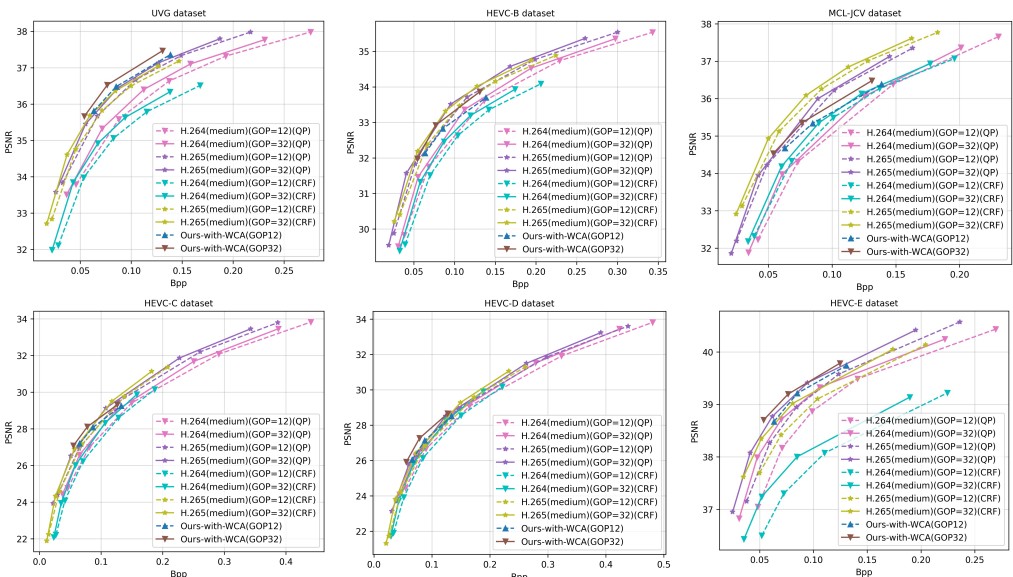

Figure 12: Rate-distortion comparison using PSNR as the metric.

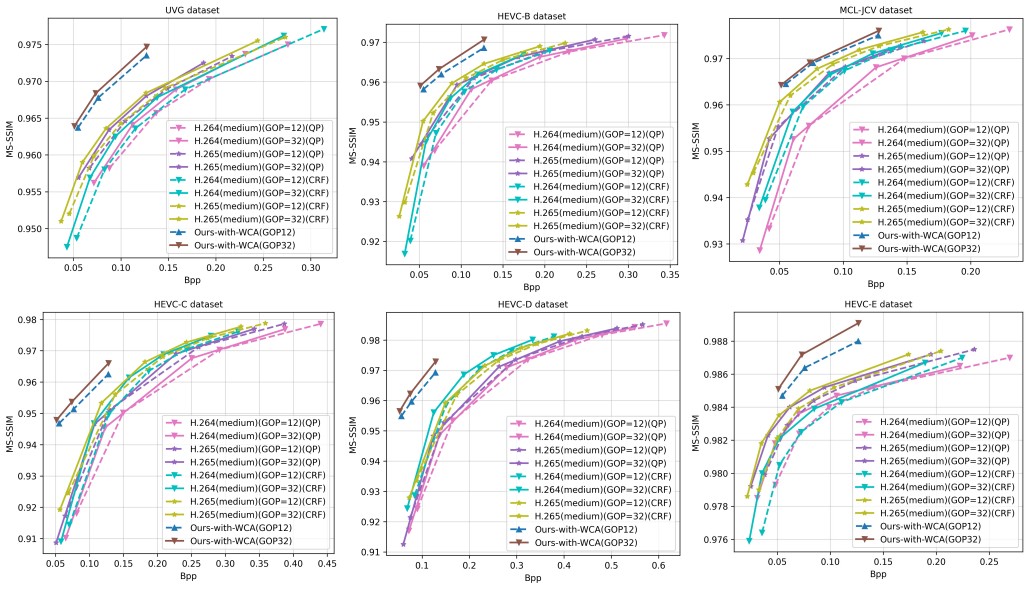

Figure 13: Rate-distortion comparison using MS-SSIM as the metric. Our model is fine-tuned for MS-SSIM.

Table 7: The detailed settings of x264 and x265.

| **x264-CRF** | **x265-CRF** |
|---|---|
| ffmpeg | ffmpeg |
| -pix_fmt yuv420p | -pix_fmt yuv420p |
| -s *widthxheight* | -s *widthxheight* |
| -i *input_file_name* | -i *input_file_name* |
| -c:v libx264 | -c:v libx265 |
| -tune zerolatency | -tune zerolatency |
| -preset medium | -preset medium |
| -x264-params | -x265-params |
| **"crf**=*qp*:keyint=*gop*:verbose=1" | **"crf**=*qp*:keyint=*gop*:verbose=1" |
| *output_video_file_name* | *output_video_file_name* |
| **x264-QP** | **x265-QP** |
| ffmpeg | ffmpeg |
| -pix_fmt yuv420p | -pix_fmt yuv420p |
| -s *widthxheight* | -s *widthxheight* |
| -i *input_file_name* | -i *input_file_name* |
| -c:v libx264 | -c:v libx265 |
| -tune zerolatency | -tune zerolatency |
| -preset medium | -preset medium |
| -x264-params | -x265-params |
| **"qp**=*qp*:keyint=*gop*:verbose=1" | **"qp**=*qp*:keyint=*gop*:verbose=1" |
| *output_video_file_name* | *output_video_file_name* |

## G  FFMPEG SETTINGS.

In our experiments, the conventional codecs H.264 and H.265 in FFmpeg are used as the anchor with *medium* preset. In current methods(Lu et al., 2019; Li et al., 2021a; 2022), *qp* and *crf* are commonly used for compressing video with quality control, while it is known that *crf* can often achieve the maximum compression efficiency [1]. Therefore, we provide separate configurations for experiments using *qp* and *crf*. The detailed settings of x264 and x265 are shown in Table 7.

---

[1] `https://trac.ffmpeg.org/wiki/Encode/H.264`

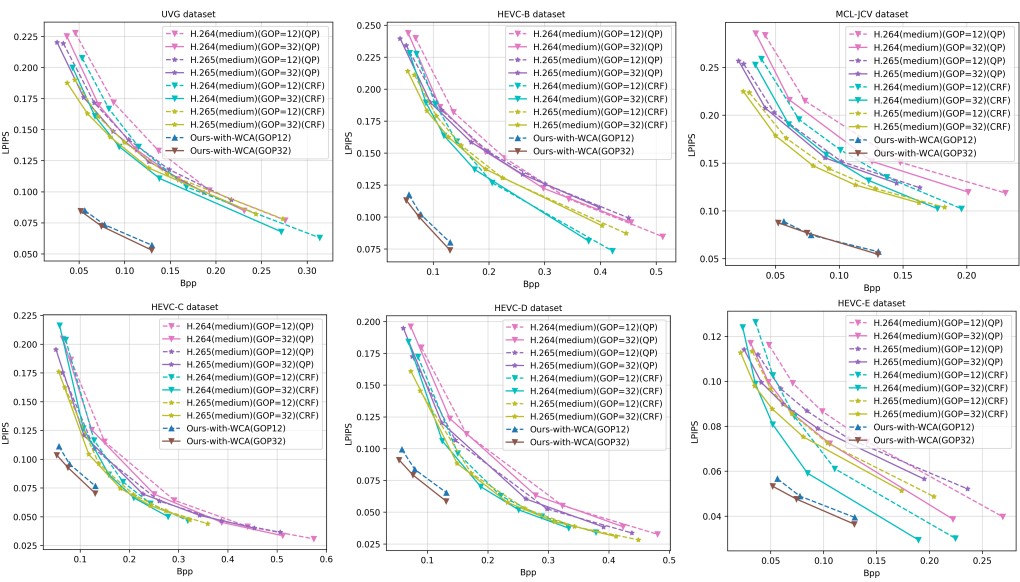

Figure 14: Rate-distortion comparison using LPIPS as the metric. Our model is fine-tuned for LPIPS.

