# OpenReview forum: "Effortless Cross-Platform Video Codec: A Codebook-Based Method"
_ICLR.cc/2024/Conference — Submitted to ICLR 2024_

### Official Review · Reviewer_g8nh · 2023-10-31

**Soundness:** 3 good
**Presentation:** 3 good
**Contribution:** 2 fair
**Rating:** 5
**Confidence:** 4

**Summary:**

The paper discusses the performance of advanced neural video codecs in comparison to traditional codecs, highlighting the role of entropy models in achieving high compression efficiency. However, it points out that cross-platform scenarios often lead to inconsistencies in probability distribution estimations due to platform-dependent floating-point computation errors, which can hinder decoding. To address this, the paper proposes a cross-platform video compression framework based on codebooks, eliminating autoregressive entropy modeling. It also introduces a conditional cross-attention module for context alignment between frames, resulting in a more efficient framework. This approach avoids the need for distribution estimation modules, making it less dependent on consistent computations across platforms.

**Strengths:**

The structure of this paper is well organized, with clear and concise explanations of the concepts. This paper addresses a significant issue in video compression by tackling the challenges of cross-platform consistency.

**Weaknesses:**

The experimental results do not effectively validate the method's effectiveness.

1.	Why were tests conducted only on 1080p videos? To validate the method's generality, the authors should test it on videos of different resolutions. Typically, in video compression tasks, testing is done on classes like B, C, D, and E, but the authors have provided results for only class B. To ensure the method's effectiveness across various resolutions, the authors should supplement the results on these different video classes. Additionally, what issues arise from center-cropping input images to multiples of 128? Why not consider the original image dimensions?

2.	The proposed method is trained on V100 FP32 and tested on both V100 FP32 and P40 FP16. I agree that different precision platforms can impact results. However, when testing with the lower precision platform, it's customary to convert FP32 to FP16 for comparison. Have the authors provided such comparisons, particularly with traditional methods like H.265 and H.264?

3.	How was the codebook generated? It is mentioned that Zhu et al.'s method is used. Could they provide specific details on the training process, or did they directly employ pre-trained codebook? If that, how can we guarantee that this codebook can adapt to the testing dataset?

4.	Why wasn't bitrate considered during training? Would joint optimization potentially yield better results?

5.	Could the authors provide an explanation for the design of codebook sizes as mentioned here? “While for predicted frames, we use three different groups of codebook sizes to achieve different video compression ratios as {8192, 2048, 512}, {64, 2048, 512} and {8, 2048, 512}.”

6.	I believe that using codebooks for compression is not only cross-platform but also an efficient approach. Why haven't the authors compared their method with more recent approaches like VVC and DCVC-DC?

7.	The authors used GOP=12 for H.264 and H.265 configurations, but the model in the paper used GOP=32. Setting H.264/H.265 configurations to GOP=32 would ensure a fairer comparison.

8.	H.264 and H.265 compression configurations seem a little different from the common test configuration. Could you provide specific instructions for H.264 and H.265 compression configurations?

9.	In Sec 2.1, “Consequently, To avoid….” here 'T' should be lowercase.

**Questions:**

Please refer to the Weaknesses section.

---

> ### Author Response · Authors · 2023-11-20
> **For Reviewer g8nh (1/4)**
>
> > 1. Why were tests conducted only on 1080p videos? To validate the method's generality, the authors should test it on videos of different resolutions. Typically, in video compression tasks, testing is done on classes like B, C, D, and E, but the authors have provided results for only class B. To ensure the method's effectiveness across various resolutions, the authors should supplement the results on these different video classes.
> > 2. The authors used GOP=12 for H.264 and H.265 configurations, but the model in the paper used GOP=32. Setting H.264/H.265 configurations to GOP=32 would ensure a fairer comparison.
>
> ***Re:*** To further demonstrate the effectiveness of our method, we conduct experiments on additional datasets with different resolutions, including HEVC Class C (480P), D (240P), and E (720P). Beyond PSNR and MS-SSIM, we introduced the perceptual evaluation metric LPIPS[1] as a supplementary experiment. In addition, we compare the performance of different GOP sizes (e.g., 12 and 32), which are commonly used in present methods[2,3,4,5,6]. Furthermore, for H.264 and H.265, we analyze the impact brought by quality control parameters (e.g., *qp* and *crf*). The detailed FFMPEG settings are shown in [Table 7](https://drive.google.com/file/d/1Ymp0lZ-a7s0JGbZy5LHyWFF8WVcxd4jK/view?usp=sharing) in the appendix of our paper.
>
> Finally, we conduct experiments on data of various resolutions and obtain rate-distortion curves for PSNR, MS-SSIM, and LPIPS, as shown in [Figure 12](https://drive.google.com/file/d/12wqNVLGpkB6RnFg0COiZJ6cclCTmjeil/view?usp=sharing), [Figure 13](https://drive.google.com/file/d/1baGqDeCr9PhDX-B71xQ2taZvbdvBIVLD/view?usp=sharing), and [Figure 14](https://drive.google.com/file/d/1O8G7a4CONyq9F6juuiQPWoovwMGqXHy8/view?usp=sharing) in the appendix of paper, respectively.
>
> As shown in Figures 13 and 14, our method significantly outperforms the best H.265 results in terms of both MS-SSIM and LPIPS. As shown in Figure 12, our method can partially outperform the H.265 results in terms of PSNR. It is worth noting that for the MCL-JCV dataset, our method presents different degrees of degradation for different metrics. The main reason for this is the presence of slight moving objects in the MCL-JCV dataset, which is discussed in [Figure 8](https://drive.google.com/file/d/1W_aAIVrQRZDiWAFHUYkCqsB_6I_xSbSH/view?usp=sharing) of our paper. And our method uses a fixed number of indices for video compression, which inevitably wastes indices in videos with high redundancy, resulting in higher bitrates for the same reconstruction quality, as shown in [Table 4](https://drive.google.com/file/d/1fNuULtK_7an9yg-PnXVaY7UI9fhy12Vw/view?usp=sharing) of our paper. This is also a direction that we need to explore in the future.
>
> More than that, we have deployed our model on mobile devices(e.g., iPhone 15 Pro) for practical validation, and the [demo video](https://drive.google.com/file/d/1F_6DOe8Rz_tIxRvROxjzbyJA9YppY1jX/view?usp=drive_link) fully demonstrate the advantages of our approach in terms of compression performance and decoding efficiency.
>
>
> > Additionally, what issues arise from center-cropping input images to multiples of 128? Why not consider the original image dimensions?
>
> ***Re:*** As shown in the model architecture in [Figure 6](https://drive.google.com/file/d/1-V4UjzXD2ih_z1BVbB8BRi-IQrIf2CW9/view?usp=sharing) of the appendix, we obtain $\mathbf{y}$ by 8× downsampling. As shown in [Figure 4](https://drive.google.com/file/d/1e-hnkIWQoXTPLMmglUX0A1XD3WJ8magw/view?usp=sharing) of our paper, in the process of multi-stage multi-codebook VQ on $\mathbf{y}$, an additional 8× downsampling is required. For $\mathbf{y}^{''}_d$, it also needs to undergo a multi-codebook quantization, which includes a 2× downsampling. Therefore, our smallest scale feature needs to undergo 128× downsampling, which is why we crop to ensure that the input image shape is divisible by 128. Certainly, there may be an optimal choice, which is to perform padding before cropping. However, for alignment with performance calculation and efficiency evaluation, we chose the cropping approach.

---

> ### Author Response · Authors · 2023-11-20
> **For Reviewer g8nh (2/4)**
>
> > The proposed method is trained on V100 FP32 and tested on both V100 FP32 and P40 FP16. I agree that different precision platforms can impact results. However, when testing with the lower precision platform, it's customary to convert FP32 to FP16 for comparison. Have the authors provided such comparisons, particularly with traditional methods like H.265 and H.264?
>
>
> ***Re:*** To our best knowledge, existing high-performance AI **video codecs** [2,3,4,5,6,9] are all based on **entropy model**, the core principle of which is to transmit data at a lower bitrate by more accurate estimation of the probability distribution, i.e., the output of the entropy model. To accurately transmit data using arithmetic coding, it is necessary to ensure that the probability distributions used during compression and decompression are identical. However, in practice, the entropy model runs on **different platforms**. At this point, when running the entropy model in floating point operations (either half-precision or single-precision, i.e., FP16 or FP32), there is usually a slight deviation, caused by unavoidable system errors, which leads to decoding errors.
>
> In summary, the cross-platform problem is not caused by low precision such as FP16, but by the fact that the entropy model needs to be identical on the encoding and decoding side. This is the motivation for our proposal of a codebook-based approach to alternative entropy model-based approaches.
>
> We focus on how to enable video codecs to be cross-platform without the need for quantization. Therefore, we validate the cross-platform capability of our model under varying platforms with different precisions. Existing entropy model-based methods face serious cross-platform issues even when using single precision floating points (i.e., FP32) on different machines. The cross-platform problem analysis for this phenomenon is presented in [Table 2](https://drive.google.com/file/d/1kZKcof-vRrr9FXsaOnjA6eBkaeRIjIZ8/view?usp=sharing) of the paper[5].
>
> H.264 and H.265 are representative of traditional codecs, and their specification and synchronization operations are mainly performed on **integer precision**[8]. This helps ensure consistency and reproducibility of the encoding and decoding process across different hardware and software platforms. This is also why we chose H.264 and H.265, which are completely cross-platform codecs, for comparison.
>
>
> > How was the codebook generated? It is mentioned that Zhu et al.'s method is used. Could they provide specific details on the training process, or did they directly employ pre-trained codebook? If that, how can we guarantee that this codebook can adapt to the testing dataset?
>
> ***Re:*** All of our codebooks are trained from scratch. Zhu et al. [7] described in the discussion section, where they trained their model end-to-end and used the Gumbel reparameterization method to train the codebook. We adopt this strategy and train our video codec, including all codebooks trained from scratch, on the training set. Ultimately, we evaluate the performance of the model on the test datasets.
>
>
> > Why wasn't bitrate considered during training? Would joint optimization potentially yield better results?
>
>
> ***Re:*** The idea you proposed is exactly our future research direction. In this paper, we did not impose any constraints on the entropy of the index. Instead, we achieve different bitrates and reconstruction qualities through manually designed codebook sizes. But in the future, we will consider designing entropy constraints for the indices of different videos to achieve content-based adaptive compression in future work. We believe that joint optimization by adding the entropy constraint of the index to the training process can improve the compression performance.
>
>
> > Could the authors provide an explanation for the design of codebook sizes as mentioned here? “While for predicted frames, we use three different groups of codebook sizes to achieve different video compression ratios as {8192, 2048, 512}, {64, 2048, 512} and {8, 2048, 512}.”
>
> ***Re:*** In paper [7], different compression ratios for images are achieved by adjusting the value of parameter $\mathbf{m}$. For video compression, due to the spatial and temporal redundancy in predicted frames, the bits per pixel (bpp) of predicted frames are typically smaller than that of keyframes. Based on the keyframe configuration in our proposed method, there are two options for adjusting the bpp of predicted frames: reducing $\mathbf{m}$ or reducing the size of the codebook. Since $\mathbf{m}$ is already set to 2, there is limited margin for adjustment. Therefore, we choose to reduce the size of the codebook to adjust the bitrate of the predicted frame. Certainly, the codebook size and $\mathbf{m}$ are hyperparameters, and we can have many different adjustment strategies, which is also one of our future research areas.

---

> ### Author Response · Authors · 2023-11-20
> **For Reviewer g8nh (3/4)**
>
> > I believe that using codebooks for compression is not only cross-platform but also an efficient approach. Why haven't the authors compared their method with more recent approaches like VVC and DCVC-DC?
>
> ***Re:*** Yes, you are absolutely correct. The codebook-based method not only solves the cross-platform issue but is also relatively efficient. We also validate VVC and SOTA AI video codecs on the same machine, and the experimental results in Table S1 demonstrate the high efficiency of our method.
>
> The essence of this article is to propose a novel codebook-based framework as a replacement for the existing entropy model-based framework, which can address cross-platform issues. Due to the absence of autoregressive modeling and optical flow alignment, we can design an extremely minimalist framework that can greatly benefit computational efficiency. Our previous experiments focus on demonstrating the feasibility of the new framework and its ability to address cross-platform issues, but lack efficiency validations. We hope that the supplementary information provided can help address any questions or concerns you may have.
>
> More than that, we have deployed our model on mobile devices(e.g., iPhone 15 Pro) for practical validation, and the [demo video](https://drive.google.com/file/d/1F_6DOe8Rz_tIxRvROxjzbyJA9YppY1jX/view?usp=drive_link) fully demonstrate the advantages of our approach in terms of decoding efficiency.
>
> **Table S1** : Efficiency comparison of different methods for the same machine and same input shape.
> | Method | Encoding time | Decoding time |
> | :-----:| ----: | ----: |
> | VVC(VTM-22.2) | 647300 ms | 161 ms |
> | DCVC(autoregressive) [2] | 6442 ms | 33865 ms |
> | DCVC-TCM [10] | 683 ms | 428 ms |
> | DCVC-HEM [3] | 680 ms | 333 ms |
> | DCVC-DC [4] | 438 ms | 225 ms |
> | WCA-based-4 (Ours) | 122 ms | 229 ms |
> | light-decoder (Ours) | 122 ms | 35 ms |
> | light-decoder (on iPad Pro) | -- | 25 ms |
> | light-decoder (on iPhone 15 Pro) | -- | 26 ms |
> | light-decoder (on iPhone 13 Pro) | -- | 31 ms |
>
>
> > H.264 and H.265 compression configurations seem a little different from the common test configuration. Could you provide specific instructions for H.264 and H.265 compression configurations?
>
> ***Re:*** In our experiments, the conventional codecs H.264 and H.265 in FFmpeg are used as the anchor with *medium* preset and Constant Rate Factor (CRF) mode. In current methods [2, 3, 9], *qp* and *crf* are commonly used for compressing video with quality control, while it is known that *crf* can often achieve the maximum compression efficiency(https://trac.ffmpeg.org/wiki/Encode/H.264). Therefore, we provide separate configurations for experiments using *qp* (i.e., for quick encoding) and *crf* (for achieving the best possible quality), which are shown in Table 1.
>
> > In Sec 2.1, “Consequently, To avoid….” here 'T' should be lowercase.
>
> ***Re:*** Thank you for the meticulous review. We have corrected the [typo](https://drive.google.com/file/d/15S7VB6gj0O5-VvN8zOK9Vm_4RmkBPWl_/view?usp=sharing) and successfully uploaded the updated paper.

---

> ### Author Response · Authors · 2023-11-20
> **For Reviewer g8nh (4/4)**
>
> **References**
>
> [1] Richard Zhang, Phillip Isola, Alexei A Efros, Eli Shechtman, and Oliver Wang. The unreasonable effectiveness of deep features as a perceptual metric. In Proceedings of the IEEE conference on computer vision and pattern recognition, pp. 586–595, 2018.
>
> [2] Jiahao Li, Bin Li, and Yan Lu. Deep contextual video compression. Advances in Neural Information Processing Systems, 34:18114–18125, 2021a.
>
> [3] Jiahao Li, Bin Li, and Yan Lu. Hybrid spatial-temporal entropy modelling for neural video compression. In Proceedings of the 30th ACM International Conference on Multimedia, pp. 1503–1511, 2022.
>
> [4] Jiahao Li, Bin Li, and Yan Lu. Neural video compression with diverse contexts. arXiv preprint arXiv:2302.14402, 2023.
>
> [5] Kuan Tian, Yonghang Guan, Jinxi Xiang, Jun Zhang, Xiao Han, and Wei Yang. Towards real-time neural video codec for cross-platform application using calibration information. arXiv preprint arXiv:2309.11276, 2023.
>
> [6] Guo Lu, Wanli Ouyang, Dong Xu, Xiaoyun Zhang, Chunlei Cai, and Zhiyong Gao. Dvc: An end-to-end deep video compression framework. In Proceedings of the IEEE/CVF Conference on Computer Vision and Pattern Recognition, pp. 11006–11015, 2019.
>
> [7] Xiaosu Zhu, Jingkuan Song, Lianli Gao, Feng Zheng, and Heng Tao Shen. Unified multivariate gaussian mixture for efficient neural image compression. In Proceedings of the IEEE/CVF Conference on Computer Vision and Pattern Recognition, pp. 17612–17621, 2022.
>
> [8] Vivienne Sze and Detlev Marpe. Entropy coding in hevc. In High Efficiency Video Coding (HEVC) Algorithms and Architectures, pp. 209–274. Springer, 2014.
>
> [9] Guo Lu, Wanli Ouyang, Dong Xu, Xiaoyun Zhang, Chunlei Cai, and Zhiyong Gao. Dvc: An end-to-end deep video compression framework. In Proceedings of the IEEE/CVF Conference on Computer Vision and Pattern Recognition, pp. 11006–11015, 2019.
>
> [10] Xihua Sheng, Jiahao Li, Bin Li, Li Li, Dong Liu, and Yan Lu. Temporal context mining for learned video compression. IEEE Transactions on Multimedia, 2022.

---

> ### Author Response · Authors · 2023-11-23
> **Friendly Reminder: Please Review Our Updated Submission and Adjust Scores**
>
> Dear Reviewer g8nh,
>
> I hope this message finds you well. I wanted to kindly remind you that we have recently submitted our revised manuscript and addressed the comments you provided during the initial review. We appreciate the valuable feedback you have given us, and we have made every effort to improve our work based on your suggestions.
>
> As the deadline for the review process is approaching, we kindly request that you take some time to review our updated submission and adjust your evaluation scores accordingly. Your timely response will help ensure a smooth and efficient review process for all parties involved.
>
> We understand that you have a busy schedule, and we truly appreciate your time and effort in reviewing our work. If you have any further questions or require clarification on any aspect of our submission, please do not hesitate to reach out. We are more than happy to provide any additional information you may need.
>
> Thank you once again for your valuable input and for helping us improve our research. We look forward to receiving your updated review and feedback.

---

### Official Review · Reviewer_6Bc6 · 2023-11-01

**Soundness:** 3 good
**Presentation:** 3 good
**Contribution:** 2 fair
**Rating:** 5
**Confidence:** 5

**Summary:**

The paper examines video decoding errors in cross-platform scenarios. Inconsistent probability distribution estimations may arise from platform-specific floating point computation errors, which can result in decoding failures for compressed bitstreams. The paper suggests a video compression framework that employs codebooks to represent temporal and spatial redundancy. This framework utilizes a WCA-based context model instead of autoregressive modeling and optical flow alignment.

**Strengths:**

1. This paper focuses on an important and practical issue of neural video compression. And it proposes one reasonable solution.
2. The design of the multi-stage codebook and window-based cross-attention successfully replaces the common motion compensation and autoregressive modules.
3. The performance of its proposed method is acceptable, which is higher than two common traditional codecs H.265 and H.264.

**Weaknesses:**

1. The novelty of this paper may be limited as the overall framework bears resemblance to Mentzer et al.'s (2022) approach, which avoids explicit motion estimation by employing a transform-like architecture. Additionally, the use of vector quantization for compression is not new and may have been inspired by Zhu's work in image compression. Furthermore, the windows-based cross attention appears similar to SwinTransformer. Consequently, the major architecture and designs lack sufficient novelty.

2. The experimental results are unconvincing as they only provide data on 1080p videos without including other resolutions such as Class C/D/E. Moreover, the authors solely compare their approach with traditional video codecs like H.265; however, it would be beneficial to include comparisons with learned video codecs that can also be utilized on CPU platforms to reduce decoding errors. These methods should be incorporated into the study.

3. In the experiment section, it is important to set an equal Group of Pictures (GoP) size when comparing the author's model with traditional codecs. Therefore, both H.265 and H.264 GoP sizes should be set at 32 for a fair comparison. Additionally, regarding the proposed codebook method, there might be concerns about error propagation if a larger GoP size is used; thus, it would be valuable for the authors to address this issue in their discussion.

**Questions:**

How do you calculate the bitrate? And how do you compress the index in the proposed framework? The authors don't provide many details.

---

> ### Author Response · Authors · 2023-11-20
> **For Reviewer 6Bc6 (1/3)**
>
> > The novelty of this paper may be limited as the overall framework bears resemblance to Mentzer et al.'s (2022) approach, which avoids explicit motion estimation by employing a transform-like architecture. Additionally, the use of vector quantization for compression is not new and may have been inspired by Zhu's work in image compression. Furthermore, the windows-based cross attention appears similar to SwinTransformer. Consequently, the major architecture and designs lack sufficient novelty.
>
> ***Re:*** Video compression and image compression are two related but different issues. Existing codebook-based approaches [1] can only address **image compression**, whereas we focus more on **video compression**, which has more complex scenarios.
>
> In video compression, the mainstream framework is based on the **entropy model**, whose critical prediction frame model mainly consists of a motion compensation module and a context compensation module. Therefore, within this framework, Mentzer et al. proposed a novel transformer-based entropy model. This model is capable of estimating the probability distribution of the current frame without the need for motion compensation [2], while still operating within the framework of **entropy model**-based methods.
>
> **In contrast**, we propose a new **codebook-based** framework rather than an entropy model-based framework for video compression. First, our proposed method eliminates the need for entropy modeling and achieves cross-platform compatibility through a codebook strategy. Furthermore, we propose a **WCA-based** context model that **avoids** the need for motion compensation and autoregressive components. Due to the absence of autoregressive modeling and optical flow alignment, we can design an extremely minimalist framework that can greatly benefit computational efficiency.
>
> Additionally, SwinTransformer [3] is a **self-attentive** framework, but our goal is to achieve more adequate information fusion between successive frames. Therefore, we propose a context model based on **cross-attention**. To improve the efficiency of our model, we further introduce a WCA(window-based cross-attention)-based context model.
>
> > 1. The experimental results are unconvincing as they only provide data on 1080p videos without including other resolutions such as Class C/D/E.
> > 2. In the experiment section, it is important to set an equal Group of Pictures (GoP) size when comparing the author's model with traditional codecs. Therefore, both H.265 and H.264 GoP sizes should be set at 32 for a fair comparison.
>
> ***Re:*** To further demonstrate the effectiveness of our method, we conduct experiments on additional datasets with different resolutions, including HEVC Class C (480P), D (240P), and E (720P). Beyond PSNR and MS-SSIM, we introduced the perceptual evaluation metric LPIPS[4] as a supplementary experiment. In addition, we compare the performance of different GOP sizes (e.g., 12 and 32), which are commonly used in present methods[5,6,7,8,9]. Furthermore, for H.264 and H.265, we analyze the impact brought by quality control parameters (e.g., *qp* and *crf*). The detailed FFMPEG settings are shown in [Table 7](https://drive.google.com/file/d/1Ymp0lZ-a7s0JGbZy5LHyWFF8WVcxd4jK/view?usp=sharing) in the appendix of our paper.
>
> Finally, we conduct experiments on data of various resolutions and obtain rate-distortion curves for PSNR, MS-SSIM, and LPIPS, as shown in [Figure 12](https://drive.google.com/file/d/12wqNVLGpkB6RnFg0COiZJ6cclCTmjeil/view?usp=sharing), [Figure 13](https://drive.google.com/file/d/1baGqDeCr9PhDX-B71xQ2taZvbdvBIVLD/view?usp=sharing), and [Figure 14](https://drive.google.com/file/d/1O8G7a4CONyq9F6juuiQPWoovwMGqXHy8/view?usp=sharing) in the appendix of paper, respectively.
>
> As shown in Figures 13 and 14, our method significantly outperforms the best H.265 results in terms of both MS-SSIM and LPIPS. As shown in Figure 12, our method can partially outperform the H.265 results in terms of PSNR. It is worth noting that for the MCL-JCV dataset, our method presents different degrees of degradation for different metrics. The main reason for this is the presence of slight moving objects in the MCL-JCV dataset, which is discussed in [Figure 8](https://drive.google.com/file/d/1W_aAIVrQRZDiWAFHUYkCqsB_6I_xSbSH/view?usp=sharing) of our paper. And our method uses a fixed number of indices for video compression, which inevitably wastes indices in videos with high redundancy, resulting in higher bitrates for the same reconstruction quality, as shown in [Table 4](https://drive.google.com/file/d/1fNuULtK_7an9yg-PnXVaY7UI9fhy12Vw/view?usp=sharing) of our paper. This is also a direction that we need to explore in the future.

---

> ### Author Response · Authors · 2023-11-20
> **For Reviewer 6Bc6 (2/3)**
>
> More than that, we have deployed our model on mobile devices(e.g., iPhone 15 Pro) for practical validation, and the [demo video](https://drive.google.com/file/d/1F_6DOe8Rz_tIxRvROxjzbyJA9YppY1jX/view?usp=drive_link) fully demonstrate the advantages of our approach in terms of compression performance and decoding efficiency.
>
> > Moreover, the authors solely compare their approach with traditional video codecs like H.265; however, it would be beneficial to include comparisons with learned video codecs that can also be utilized on CPU platforms to reduce decoding errors. These methods should be incorporated into the study.
>
> ***Re:*** To our best knowledge, existing high-performance AI **video codecs** [2,5,6,7,8,9] are all based on **entropy model**, the core principle of which is to transmit data at a lower bitrate by more accurate estimation of the probability distribution, i.e., the output of the entropy model. To accurately transmit data using arithmetic coding, it is necessary to ensure that the probability distributions used during compression and decompression are identical. However, in practice, the entropy model runs on **different platforms**. At this point, when running the entropy model in floating point operations (either half-precision or single-precision, i.e., FP16 or FP32), there is usually a slight deviation, caused by unavoidable system errors, which leads to decoding errors.
>
> Currently, without considering model quantization, only the calibration information transmission method has implemented a cross-platform video codec model [8]. They developed a plugin for entropy model-based video codec, which obtains cross-platform capabilities by transmitting additional bitstream. However, this method can only partially solve the cross-platform problem. We compare the differences in cross-platform capabilities between these two methods in Table S1.
>
> In addition, regarding the use of CPU to reduce the decoding errors of entropy model-based AI video codecs, a detailed experimental analysis has been conducted in [Table 2](https://drive.google.com/file/d/1kZKcof-vRrr9FXsaOnjA6eBkaeRIjIZ8/view?usp=drive_link) in paper [8]. Any different computing system, including different hardware, software, and the use of different CPUs for encoding and decoding as you mentioned, will introduce system errors, resulting in decoding errors in entropy model-based methods. This is also why we can only compare the performance with traditional codecs like H.264 and H.265, which are fully cross-platform codecs.
>
> **Table S1** : Decompression failure rates (lower better) across different platforms.
> | Encoder precision | CPU1 FP32 | CPU1 FP32 | V100 FP32 | V100 FP32 |  V100 FP32 | V100 FP32 | V100 FP32 |
> | :-----:| :----: | :----: | :----: | :----: | :----:| :----:| :----:|
> | **Decoder precision** | **CPU1 FP32** | **CPU2 FP32** | **V100 FP32** | **V100 FP16** |  **P40 FP32** | **P40 FP16** | **iPhone ANE FP16** |
> | DCVC-HEM[6] | 0% | 94.8% | 0% | 100% | 100% | 100% | 100% |
> | CIT[8] | 0% | 0% | 0% | 100% | 0% | 100% | 100% |
> | Ours | 0% | 0% | 0% | 0% | 0% | 0% | 0% |
>
>
> > Additionally, regarding the proposed codebook method, there might be concerns about error propagation if a larger GoP size is used; thus, it would be valuable for the authors to address this issue in their discussion.
>
> ***Re:*** First, existing entropy-based methods require consistent probability distribution estimation, which is difficult to guarantee in cross-platform scenarios. Therefore, entropy model-based methods **cannot** achieve cross-platform compatibility. In this paper, we propose a codebook-based video codec without entropy modeling, which **can** achieve cross-platform compatibility.
>
> Of course, like all neural networks, different platforms can lead to error propagation. This can be optimized by choosing operators with smaller errors or other engineering strategies. According to our experiments, setting the GOP size to 32 (the same as the existing SOTA method[6,7]) does not cause significant cumulative errors in our model in all test environments (e.g., iPad Pro ANE, iPad Pro GPU, iPhone 15 Pro ANE, iPhone 15 Pro GPU, etc).

---

> ### Author Response · Authors · 2023-11-20
> **For Reviewer 6Bc6 (3/3)**
>
> > How do you calculate the bitrate? And how do you compress the index in the proposed framework? The authors don't provide many details.
>
> ***Re:*** I'm sorry for any inconvenience I may have caused you. The paper discusses the upper bound of bitrate required for transmitting indices of multi-stage multi-codebook[1]. For example, if the codebook size is **8192**, transmitting an index requires **13** bits, since $log_{2}8192=13$. If we set a multi-codebook size of 2, transmitting a code requires **26** bits, which is 2 times the number of bits needed for a single codebook index (i.e., $2*13=26$ bits). The latent shape is $H/16 \times W/16$, where $H$ and $W$ are the shape of the input image. Therefore, transmitting one frame requires $H/16 \times W/16 \times 26 / (H \times W) = 0.1015$ bits per pixel (bpp).
>
> Certainly, we can calculate the probability of different indices appearing in the training set, and then take advantage of arithmetic coding to compress the indexes with higher probabilities into smaller bits, which reduces the final bpp required for transmission. **For all experiments in our paper**, the bitstream that can be transmitted is obtained, and the bpp is calculated based on it.
>
> **References**
>
> [1] Xiaosu Zhu, Jingkuan Song, Lianli Gao, Feng Zheng, and Heng Tao Shen. Unified multivariate gaussian mixture for efficient neural image compression. In Proceedings of the IEEE/CVF Conference on Computer Vision and Pattern Recognition, pp. 17612–17621, 2022.
>
> [2] Fabian Mentzer, George Toderici, David Minnen, Sung-Jin Hwang, Sergi Caelles, Mario Lucic, and Eirikur Agustsson. Vct: A video compression transformer. arXiv preprint arXiv:2206.07307, 2022.
>
> [3] Ze Liu, Yutong Lin, Yue Cao, Han Hu, Yixuan Wei, Zheng Zhang, Stephen Lin, and Baining Guo. Swin transformer: Hierarchical vision transformer using shifted windows. In Proceedings of the IEEE/CVF international conference on computer vision, pp. 10012–10022, 2021.
>
> [4] Richard Zhang, Phillip Isola, Alexei A Efros, Eli Shechtman, and Oliver Wang. The unreasonable effectiveness of deep features as a perceptual metric. In Proceedings of the IEEE conference on computer vision and pattern recognition, pp. 586–595, 2018.
>
> [5] Jiahao Li, Bin Li, and Yan Lu. Deep contextual video compression. Advances in Neural Information Processing Systems, 34:18114–18125, 2021a.
>
> [6] Jiahao Li, Bin Li, and Yan Lu. Hybrid spatial-temporal entropy modelling for neural video compression. In Proceedings of the 30th ACM International Conference on Multimedia, pp. 1503–1511, 2022.
>
> [7] Jiahao Li, Bin Li, and Yan Lu. Neural video compression with diverse contexts. arXiv preprint arXiv:2302.14402, 2023.
>
> [8] Kuan Tian, Yonghang Guan, Jinxi Xiang, Jun Zhang, Xiao Han, and Wei Yang. Towards real-time neural video codec for cross-platform application using calibration information. arXiv preprint arXiv:2309.11276, 2023.
>
> [9] Guo Lu, Wanli Ouyang, Dong Xu, Xiaoyun Zhang, Chunlei Cai, and Zhiyong Gao. Dvc: An end-to-end deep video compression framework. In Proceedings of the IEEE/CVF Conference on Computer Vision and Pattern Recognition, pp. 11006–11015, 2019.

---

> ### Author Response · Authors · 2023-11-23
> **Friendly Reminder: Please Review Our Updated Submission and Adjust Scores**
>
> Dear Reviewer 6Bc6,
>
> I hope this message finds you well. I wanted to kindly remind you that we have recently submitted our revised manuscript and addressed the comments you provided during the initial review. We appreciate the valuable feedback you have given us, and we have made every effort to improve our work based on your suggestions.
>
> As the deadline for the review process is approaching, we kindly request that you take some time to review our updated submission and adjust your evaluation scores accordingly. Your timely response will help ensure a smooth and efficient review process for all parties involved.
>
> We understand that you have a busy schedule, and we truly appreciate your time and effort in reviewing our work. If you have any further questions or require clarification on any aspect of our submission, please do not hesitate to reach out. We are more than happy to provide any additional information you may need.
>
> Thank you once again for your valuable input and for helping us improve our research. We look forward to receiving your updated review and feedback.

---

### Official Review · Reviewer_z95x · 2023-11-04

**Soundness:** 3 good
**Presentation:** 3 good
**Contribution:** 3 good
**Rating:** 8
**Confidence:** 4

**Summary:**

1. In this paper authros proposed a cross-platform videocompression framework based on codebooks, which avoids autoregressive entropy
modeling and achieves video compression by transmitting the index sequence of the codebooks.

**Strengths:**

1.Novelity of the paper is good.
2. Encourging results.

**Weaknesses:**

1.Related work should be updated.

**Questions:**

1.Include the recent papers in references.
2.Mention the future scope.
3.Detailed analysis of figure 5 results is required.

---

> ### Author Response · Authors · 2023-11-20
> **For Reviewer z95x (1/2)**
>
> > Related work should be updated.
> > Include the recent papers in references.
>
> ***Re:*** Yes, there is a new interesting direction using the generative model for video or image compression [1,2]. There are also methods focusing on compression efficiency [3,4]. We have updated the related works. If there is still interesting work that we have neglected, we will be happy that the reviewer can point it out for us.
>
>
> > Mention the future scope.
>
> ***Re:*** As mentioned in the conclusions, we will consider designing entropy constraints for the indices of different videos to achieve content-based adaptive compression in future work. In addition, several reviewers have mentioned the potential efficiency benefits of the codebook-based approach, which we are currently testing on mobile devices. As shown in Table S1, the experimental results up to this point indicate significant efficiency advantages of our approach.
>
> **Table S1** : Efficiency comparison of different methods for the same machine and same input shape.
> | Method | Encoding time | Decoding time |
> | :-----:| ----: | ----: |
> | VVC(VTM-22.2) | 647300 ms | 161 ms |
> | DCVC(autoregressive) [6] | 6442 ms | 33865 ms |
> | DCVC-TCM [11] | 683 ms | 428 ms |
> | DCVC-HEM [7] | 680 ms | 333 ms |
> | DCVC-DC [8] | 438 ms | 225 ms |
> | WCA-based-4 (Ours) | 122 ms | 229 ms |
> | light-decoder (Ours) | 122 ms | 35 ms |
> | light-decoder (on iPad Pro) | -- | 25 ms |
> | light-decoder (on iPhone 15 Pro) | -- | 26 ms |
> | light-decoder (on iPhone 13 Pro) | -- | 31 ms |
>
> > Detailed analysis of figure 5 results is required.
>
> ***Re:*** To further demonstrate the effectiveness of our method, we conduct experiments on additional datasets with different resolutions, including HEVC Class C (480P), D (240P), and E (720P). Beyond PSNR and MS-SSIM, we introduced the perceptual evaluation metric LPIPS[5] as a supplementary experiment. In addition, we compare the performance of different GOP sizes (e.g., 12 and 32), which are commonly used in present methods[6,7,8,9,10,11]. Furthermore, for H.264 and H.265, we analyze the impact brought by quality control parameters (e.g., *qp* and *crf*). The detailed FFMPEG settings are shown in [Table 7](https://drive.google.com/file/d/1Ymp0lZ-a7s0JGbZy5LHyWFF8WVcxd4jK/view?usp=sharing) in the appendix of our paper.
>
> Finally, we conduct experiments on data of various resolutions and obtain rate-distortion curves for PSNR, MS-SSIM, and LPIPS, as shown in [Figure 12](https://drive.google.com/file/d/12wqNVLGpkB6RnFg0COiZJ6cclCTmjeil/view?usp=sharing), [Figure 13](https://drive.google.com/file/d/1baGqDeCr9PhDX-B71xQ2taZvbdvBIVLD/view?usp=sharing), and [Figure 14](https://drive.google.com/file/d/1O8G7a4CONyq9F6juuiQPWoovwMGqXHy8/view?usp=sharing) in the appendix of paper, respectively.
>
> As shown in Figures 13 and 14, our method significantly outperforms the best H.265 results in terms of both MS-SSIM and LPIPS. As shown in Figure 12, our method can partially outperform the H.265 results in terms of PSNR. It is worth noting that for the MCL-JCV dataset, our method presents different degrees of degradation for different metrics. The main reason for this is the presence of slight moving objects in the MCL-JCV dataset, which is discussed in [Figure 8](https://drive.google.com/file/d/1W_aAIVrQRZDiWAFHUYkCqsB_6I_xSbSH/view?usp=sharing) of our paper. And our method uses a fixed number of indices for video compression, which inevitably wastes indices in videos with high redundancy, resulting in higher bitrates for the same reconstruction quality, as shown in [Table 4](https://drive.google.com/file/d/1fNuULtK_7an9yg-PnXVaY7UI9fhy12Vw/view?usp=sharing) of our paper. This is also a direction that we need to explore in the future.
>
> More than that, we have deployed our model on mobile devices(e.g., iPhone 15 Pro) for practical validation, and the [demo video](https://drive.google.com/file/d/1F_6DOe8Rz_tIxRvROxjzbyJA9YppY1jX/view?usp=drive_link) fully demonstrate the advantages of our approach in terms of compression performance and decoding efficiency.

---

> ### Author Response · Authors · 2023-11-20
> **For Reviewer z95x (2/2)**
>
> **References**
>
> [1] Ruihan Yang, Yibo Yang, Joseph Marino, and Stephan Mandt. Insights from generative modeling for neural video compression. IEEE Transactions on Pattern Analysis and Machine Intelligence, 2023.
>
> [2] Ruihan Yang and Stephan Mandt. Lossy image compression with conditional diffusion models. arXiv preprint arXiv:2209.06950, 2022.
>
> [3] Ho Man Kwan, Ge Gao, Fan Zhang, Andrew Gower, and David Bull. Hinerv: Video compression with hierarchical encoding based neural representation. arXiv preprint arXiv:2306.09818, 2023.
>
> [4] Muhammad Salman Ali, Yeongwoong Kim, et al. Towards efficient image compression without autoregressive models. Advances in Neural Information Processing Systems, 2023.
>
> [5] Richard Zhang, Phillip Isola, Alexei A Efros, Eli Shechtman, and Oliver Wang. The unreasonable effectiveness of deep features as a perceptual metric. In Proceedings of the IEEE conference on computer vision and pattern recognition, pp. 586–595, 2018.
>
> [6] Jiahao Li, Bin Li, and Yan Lu. Deep contextual video compression. Advances in Neural Information Processing Systems, 34:18114–18125, 2021a.
>
> [7] Jiahao Li, Bin Li, and Yan Lu. Hybrid spatial-temporal entropy modelling for neural video compression. In Proceedings of the 30th ACM International Conference on Multimedia, pp. 1503–1511, 2022.
>
> [8] Jiahao Li, Bin Li, and Yan Lu. Neural video compression with diverse contexts. arXiv preprint arXiv:2302.14402, 2023.
>
> [9] Kuan Tian, Yonghang Guan, Jinxi Xiang, Jun Zhang, Xiao Han, and Wei Yang. Towards real-time neural video codec for cross-platform application using calibration information. arXiv preprint arXiv:2309.11276, 2023.
>
> [10] Guo Lu, Wanli Ouyang, Dong Xu, Xiaoyun Zhang, Chunlei Cai, and Zhiyong Gao. Dvc: An end-to-end deep video compression framework. In Proceedings of the IEEE/CVF Conference on Computer Vision and Pattern Recognition, pp. 11006–11015, 2019.
>
> [11] Xihua Sheng, Jiahao Li, Bin Li, Li Li, Dong Liu, and Yan Lu. Temporal context mining for learned video compression. IEEE Transactions on Multimedia, 2022.

---

### Official Review · Reviewer_iHcA · 2023-11-05

**Soundness:** 3 good
**Presentation:** 3 good
**Contribution:** 2 fair
**Rating:** 6
**Confidence:** 4

**Summary:**

This paper presents a video compression framework that facilitates cross-platform compatibility by avoiding the need for entropy modeling in probability distribution estimation. To achieve this, this paper leverages codebook-based approaches and proposes window-based cross-attention that avoids the use of optical flow for context alignments. The paper demonstrates the effectiveness of the proposed method compared to existing traditional video codecs.

**Strengths:**

- The paper is well-written and flows smoothly.
- The motivation for the proposed method seems intriguing in the context of neural video compression.

**Weaknesses:**

1. **More RD-performance comparison with existing neural video compression methods**: The paper primarily made a comparison with traditional codecs like H.264 and H.265. While Figure 1 demonstrates the artifacts induced by entropy models in cross-platform settings, the paper does not conclusively establish if this issue is prevalent across all neural video compression methods. To strengthen the paper's claims, a more comprehensive RD-performance comparison with a variety of neural methods in similar cross-platform scenarios is necessary. Moreover, the paper should include both qualitative and quantitative results. Given that metrics like PSNR and MS-SSIM might not fully represent the artifacts seen in decompressed frames, the inclusion of perceptual similarity metrics such as LPIPS [1] or FLIP [2] is recommended for a more comprehensive evaluation.

2. **Decoding efficiency**: The paper should also report on decoding efficiency relative to existing neural video compression methods, extending beyond the ablation studies of the proposed modules.

3. **More consideration of cross-platform scenario when encoding/decoding**: The paper presents the scenarios where encoding is done using an NVIDIA Tesla V100 and decoding with a Tesla P40, as detailed in Table 5 of the Appendix. However, this scenario is limited. More examination of diverse scenarios including a broader range of encoding/decoding machines should be included to verify the robustness of the proposed method regardless of the systematic errors across different platforms.

4. **More comparison with existing calibration methods**:
The paper should incorporate a detailed comparison with existing calibration methods. This is crucial for contextualizing the results presented in Table 2 and Table 5. The inclusion of performance comparison with existing calibration methods will significantly enhance the paper's comprehensiveness and the validity of its conclusions.



[1] Zhang et al., The Unreasonable Effectiveness of Deep Features as a Perceptual Metric

[2] Andersson et al., A Difference Evaluator for Alternating Images

**Questions:**

The proposed WCA-based context model is intriguing. However, the presented modules, including the codebook approach and the keyframe and prediction frame categorization, appear as already well-established techniques in the field of neural data compression. While the focus on cross-platform scenarios is an interesting aspect that underscores the effectiveness of your proposed method, it also requires a deeper investigation to distinguish this paper from existing methods. Therefore, to strengthen the claim of the paper, a more comprehensive comparison as mentioned in the weakness section seems crucial. Without them, my evaluation of the paper may be lower.

---

> ### Author Response · Authors · 2023-11-20
> **For Reviewer iHcA (1/3)**
>
> > The paper primarily made a comparison with traditional codecs like H.264 and H.265. While Figure 1 demonstrates the artifacts induced by entropy models in cross-platform settings, the paper does not conclusively establish if this issue is prevalent across all neural video compression methods.
>
> ***Re:*** All video codecs based on entropy models face cross-platform issues.
>
> To our best knowledge, existing high-performance AI **video codecs** [4,5,6,7,8,9,10] are all based on **entropy model**, the core principle of which is to transmit data at a lower bitrate by more accurate estimation of the probability distribution, i.e., the output of the entropy model. To accurately transmit data using arithmetic coding, it is necessary to ensure that the probability distributions used during compression and decompression are identical. However, in practice, the entropy model runs on **different platforms**. At this point, when running the entropy model in floating point operations (either half-precision or single-precision, i.e., FP16 or FP32), there is usually a slight deviation, caused by unavoidable system errors, which leads to decoding errors.
>
> Entropy model-based video codecs need to ensure that the outputs of the entropy model are identical for different platforms, otherwise, the decoding error will occur. In this paper, we illustrate the decoding error using the latest checkerboard entropy model as an example in [Figure 1](https://drive.google.com/file/d/1SklD4HIYUbwMfE7035_LnTG4A9eztUCm/view?usp=sharing). The earlier autoregressive entropy models also suffer from the same issue, but the manifestation may be different. As shown in the supplement [Figure]( https://drive.google.com/file/d/1XDyZtPjz58jq2HA2yog5FXOlEvAv-L_I/view?usp=drive_link), we try to explain these two types of decoding error in more detail, hoping to solve any confusion you may have.
>
> > To strengthen the paper's claims, a more comprehensive RD-performance comparison with a variety of neural methods in similar cross-platform scenarios is necessary.
>
> ***Re:*** To the best of our knowledge, apart from model quantization methods, which are valid for all models but require a lot of engineering effort, there are currently few methods that address the cross-platform problem of video codecs. Tian et al. propose a calibration method that transmits additional information to achieve cross platform capability [1]. However, the method can partially solve the problem because it only works under the conditions of error-limited single-precision floating point calculation ( i.e., FP32). In contrast, our method can run on different floating point computing platforms, including FP32 and FP16, without any additional plug-ins. Overall, there are currently no cross-platform capable full *AI video codecs* available for comparison.
>
> To further demonstrate the effectiveness of our method, we conduct experiments on additional datasets with different resolutions, including HEVC Class C (480P), D (240P), and E (720P). Beyond PSNR and MS-SSIM, we introduced the perceptual evaluation metric LPIPS[2] as a supplementary experiment. In addition, we compare the performance of different GOP sizes (e.g., 12 and 32), which are commonly used in present methods[4,5,6,7,8]. Furthermore, for H.264 and H.265, we analyze the impact brought by quality control parameters (e.g., *qp* and *crf*). The detailed FFMPEG settings are shown in [Table 7](https://drive.google.com/file/d/1Ymp0lZ-a7s0JGbZy5LHyWFF8WVcxd4jK/view?usp=sharing) in the appendix of our paper.
>
> Finally, we conduct experiments on data of various resolutions and obtain rate-distortion curves for PSNR, MS-SSIM, and LPIPS, as shown in [Figure 12](https://drive.google.com/file/d/12wqNVLGpkB6RnFg0COiZJ6cclCTmjeil/view?usp=sharing), [Figure 13](https://drive.google.com/file/d/1baGqDeCr9PhDX-B71xQ2taZvbdvBIVLD/view?usp=sharing), and [Figure 14](https://drive.google.com/file/d/1O8G7a4CONyq9F6juuiQPWoovwMGqXHy8/view?usp=sharing) in the appendix of paper, respectively.
>
> As shown in Figures 13 and 14, our method significantly outperforms the best H.265 results in terms of both MS-SSIM and LPIPS. As shown in Figure 12, our method can partially outperform the H.265 results in terms of PSNR. It is worth noting that for the MCL-JCV dataset, our method presents different degrees of degradation for different metrics. The main reason for this is the presence of slight moving objects in the MCL-JCV dataset, which is discussed in [Figure 8](https://drive.google.com/file/d/1W_aAIVrQRZDiWAFHUYkCqsB_6I_xSbSH/view?usp=sharing) of our paper. And our method uses a fixed number of indices for video compression, which inevitably wastes indices in videos with high redundancy, resulting in higher bitrates for the same reconstruction quality, as shown in [Table 4](https://drive.google.com/file/d/1fNuULtK_7an9yg-PnXVaY7UI9fhy12Vw/view?usp=sharing) of our paper. This is also a direction that we need to explore in the future.

---

> ### Author Response · Authors · 2023-11-20
> **For Reviewer iHcA (2/3)**
>
> More than that, we have deployed our model on mobile devices(e.g., iPhone 15 Pro) for practical validation, and the [demo video](https://drive.google.com/file/d/1F_6DOe8Rz_tIxRvROxjzbyJA9YppY1jX/view?usp=drive_link) fully demonstrate the advantages of our approach in terms of compression performance and decoding efficiency.
>
> > Moreover, the paper should include both qualitative and quantitative results. Given that metrics like PSNR and MS-SSIM might not fully represent the artifacts seen in decompressed frames, the inclusion of perceptual similarity metrics such as LPIPS or FLIP is recommended for a more comprehensive evaluation.
>
> ***Re:*** Typically, AI video codecs are usually trained on PSNR and MS-SSIM[1,4,5,6,7,8,9,10]. To more adequately demonstrate the validity of our approach, we conduct additional experiments using the perceptual metric LPIPS[2], which is more commonly used. The final results are shown in [Figure 14](https://drive.google.com/file/d/1O8G7a4CONyq9F6juuiQPWoovwMGqXHy8/view?usp=sharing) in the appendix of our paper.
>
> > Decoding efficiency: The paper should also report on decoding efficiency relative to existing neural video compression methods, extending beyond the ablation studies of the proposed modules.
>
> ***Re:*** We thank the reviewer for pointing this out. We have compared the efficiency of our method with traditional VVC(VTM) and SOTA AI video codecs. The specific encoding and decoding times are shown in Table S1. Although our performance only exceeds that of H.265, our approach has significant advantages in both encoding and decoding efficiency than the AI video codecs. This will play a crucial role in future practical applications.
>
> **Table S1** : Efficiency comparison of different methods for the same machine and same input shape.
> | Method | Encoding time | Decoding time |
> | :-----:| ----: | ----: |
> | VVC(VTM-22.2) | 647300 ms | 161 ms |
> | DCVC(autoregressive) [4] | 6442 ms | 33865 ms |
> | DCVC-TCM [10] | 683 ms | 428 ms |
> | DCVC-HEM [5] | 680 ms | 333 ms |
> | DCVC-DC [6] | 438 ms | 225 ms |
> | WCA-based-4 (Ours) | 122 ms | 229 ms |
> | light-decoder (Ours) | 122 ms | 35 ms |
> | light-decoder (on iPad Pro) | -- | 25 ms |
> | light-decoder (on iPhone 15 Pro) | -- | 26 ms |
> | light-decoder (on iPhone 13 Pro) | -- | 31 ms |
>
>
> > More consideration of cross-platform scenario when encoding/decoding: The paper presents the scenarios where encoding is done using an NVIDIA Tesla V100 and decoding with a Tesla P40, as detailed in Table 5 of the Appendix. However, this scenario is limited. More examination of diverse scenarios including a broader range of encoding/decoding machines should be included to verify the robustness of the proposed method regardless of the systematic errors across different platforms.
>
> ***Re:*** First, Since our approach eliminates the dependence on the entropy model, there is inherently no need for the computations at the encoding and decoding ends to be identical. Then our codebook-based video codec is able to run cross-platform. In addition, we have conducted cross-platform experiments on more heterogeneous platforms, and the results, as shown in Table S2, are sufficient to demonstrate the advantages of our approach in cross-platform scenarios.
>
> More than that, we have deployed our model on mobile devices(e.g., iPhone 15 Pro) for practical validation, and the [demo video](https://drive.google.com/file/d/1F_6DOe8Rz_tIxRvROxjzbyJA9YppY1jX/view?usp=drive_link) fully demonstrate the advantages of our approach in terms of compression performance and decoding efficiency.
>
> **Table S2** : BD-rate calculated on different platforms by SSIM on UVG dataset.
> | Decoder <br> precision | V100 <br> FP32 | V100 <br> FP16 |  P40 <br> FP32 | P40 <br> FP16 |  iPad GPU <br> FP16 | iPad ANE <br> FP16 |
> | :-----:| :----: | :----: | :----: | :----: | :----: | :----: |
> | UVG | 0% | +0.9% | 0% | +0.9% | +0.9% | +1.0% |
>
> > More comparison with existing calibration methods: The paper should incorporate a detailed comparison with existing calibration methods. This is crucial for contextualizing the results presented in Table 2 and Table 5. The inclusion of performance comparison with existing calibration methods will significantly enhance the paper's comprehensiveness and the validity of its conclusions.
>
> ***Re:*** To our knowledge, the only existing cross-platform AI video codec is the calibration method proposed by Tian et al. [1] They developed a plugin for entropy model-based video codec, which obtains cross-platform capabilities by transmitting additional bitstream. In contrast, we propose a new codebook-based method to address the cross-platform issue in AI video codecs. Therefore, we compare our method with the calibration method in terms of cross-platform capability, and the results are shown in Table S3. It can be seen that our method has more universal cross-platform capabilities due to the differences in the framework.

---

> ### Author Response · Authors · 2023-11-20
> **For Reviewer iHcA (3/3)**
>
> **Table S3** : Decompression failure rates (lower better) across different platforms.
> | Encoder precision | CPU1 FP32 | CPU1 FP32 | V100 FP32 | V100 FP32 |  V100 FP32 | V100 FP32 | V100 FP32 |
> | :-----:| :----: | :----: | :----: | :----: | :----:| :----:| :----:|
> | **Decoder precision** | **CPU1 FP32** | **CPU2 FP32** | **V100 FP32** | **V100 FP16** |  **P40 FP32** | **P40 FP16** | **iPhone ANE FP16** |
> | DCVC-HEM[5] | 0% | 94.8% | 0% | 100% | 100% | 100% | 100% |
> | CIT[1] | 0% | 0% | 0% | 100% | 0% | 100% | 100% |
> | Ours | 0% | 0% | 0% | 0% | 0% | 0% | 0% |
>
>
> > The proposed WCA-based context model is intriguing. However, the presented modules, including the codebook approach and the keyframe and prediction frame categorization, appear as already well-established techniques in the field of neural data compression. While the focus on cross-platform scenarios is an interesting aspect that underscores the effectiveness of your proposed method, it also requires a deeper investigation to distinguish this paper from existing methods. Therefore, to strengthen the claim of the paper, a more comprehensive comparison as mentioned in the weakness section seems crucial. Without them, my evaluation of the paper may be lower.
>
> ***Re:*** Video compression and image compression are two related but different issues. Zhu et al. propose a codebook-based approach that can only address image compression [3], whereas we focus more on video compression, which has more complex scenarios. In video compression, the mainstream framework is based on entropy modeling, whose critical prediction frame model mainly consists of a motion compensation module and a context compensation module. However, the SOTA approach also requires a parallel-unfriendly autoregressive component. **In contrast**, we propose a new codebook-based video codec. First, our framework eliminates the need for entropy modeling and achieves cross-platform compatibility through a codebook strategy. Furthermore, we propose a WCA-based context model that avoids the need for motion compensation and autoregressive components. Due to the absence of autoregressive modeling and optical flow alignment, we can design an extremely minimalist framework that can greatly benefit computational efficiency.
>
> **References**
>
> [1] Kuan Tian, Yonghang Guan, Jinxi Xiang, Jun Zhang, Xiao Han, and Wei Yang. Towards real-time neural video codec for cross-platform application using calibration information. arXiv preprint arXiv:2309.11276, 2023.
>
> [2] Richard Zhang, Phillip Isola, Alexei A Efros, Eli Shechtman, and Oliver Wang. The unreasonable effectiveness of deep features as a perceptual metric. In Proceedings of the IEEE conference on computer vision and pattern recognition, pp. 586–595, 2018.
>
> [3] Xiaosu Zhu, Jingkuan Song, Lianli Gao, Feng Zheng, and Heng Tao Shen. Unified multivariate gaussian mixture for efficient neural image compression. In Proceedings of the IEEE/CVF Conference on Computer Vision and Pattern Recognition, pp. 17612–17621, 2022.
>
> [4] Jiahao Li, Bin Li, and Yan Lu. Deep contextual video compression. Advances in Neural Information Processing Systems, 34:18114–18125, 2021a.
>
> [5] Jiahao Li, Bin Li, and Yan Lu. Hybrid spatial-temporal entropy modelling for neural video compression. In Proceedings of the 30th ACM International Conference on Multimedia, pp. 1503–1511, 2022.
>
> [6] Jiahao Li, Bin Li, and Yan Lu. Neural video compression with diverse contexts. arXiv preprint arXiv:2302.14402, 2023.
>
> [7] Guo Lu, Wanli Ouyang, Dong Xu, Xiaoyun Zhang, Chunlei Cai, and Zhiyong Gao. Dvc: An end-to-end deep video compression framework. In Proceedings of the IEEE/CVF Conference on Computer Vision and Pattern Recognition, pp. 11006–11015, 2019.
>
> [8] Fabian Mentzer, George Toderici, David Minnen, Sung-Jin Hwang, Sergi Caelles, Mario Lucic, and Eirikur Agustsson. Vct: A video compression transformer. arXiv preprint arXiv:2206.07307, 2022.
>
> [9] Jinxi Xiang, Kuan Tian, and Jun Zhang. Mimt: Masked image modeling transformer for video compression. In The Eleventh International Conference on Learning Representations, 2022.
>
> [10] Xihua Sheng, Jiahao Li, Bin Li, Li Li, Dong Liu, and Yan Lu. Temporal context mining for learned video compression. IEEE Transactions on Multimedia, 2022.

---

> ### Author Response · Authors · 2023-11-23
> **Friendly Reminder: Please Review Our Updated Submission and Adjust Scores**
>
> Dear Reviewer iHcA,
>
> I hope this message finds you well. I wanted to kindly remind you that we have recently submitted our revised manuscript and addressed the comments you provided during the initial review. We appreciate the valuable feedback you have given us, and we have made every effort to improve our work based on your suggestions.
>
> As the deadline for the review process is approaching, we kindly request that you take some time to review our updated submission and adjust your evaluation scores accordingly. Your timely response will help ensure a smooth and efficient review process for all parties involved.
>
> We understand that you have a busy schedule, and we truly appreciate your time and effort in reviewing our work. If you have any further questions or require clarification on any aspect of our submission, please do not hesitate to reach out. We are more than happy to provide any additional information you may need.
>
> Thank you once again for your valuable input and for helping us improve our research. We look forward to receiving your updated review and feedback.

---

> > ### Comment · Reviewer_iHcA · 2023-11-23
> >
> > Thanks for the rebuttal, and most of my concerns are addressed. However, I am not asking why the decoding error happens, but how much the entropy-based video compression models suffer from decoding errors quantitatively. Therefore, I recommend comparing it to a variety of entropy-based video compression schemes quantitatively in the final version of the manuscript, even if they suffer from decoding errors.

---

> > > ### Author Response · Authors · 2023-11-23
> > > **Reply to Suggestions**
> > >
> > > Thank you for your suggestions. I try to understand that you want us to quantitatively compare the metrics of the image with decoding errors based on entropy modeling with the methods in this paper. However, unfortunately, the decoding error we show in Figure 1 of this paper is only the second frame of a GOP, and all subsequent frames are dependent on the previous frame for reconstruction, so the error in subsequent frames gets larger and larger. And it takes only a very small number of frames to become completely noisy. This is one of the main reasons why we cannot make quantitative comparisons.
> > >
> > > I hope my reply addresses your questions. Thanks again for your meticulous review!

---

### Author Response · Authors · 2023-11-20
**For all reviewers (1/2)**

As the performance of AI video codec continues to improve, the cross-platform issue it faces in practical applications has become increasingly critical. In this paper, we propose a codebook-based video codec to tackle this significant challenge.

We thank the reviewers for their thoughtful and constructive review of our manuscript. We are greatly encouraged to hear the reviewers' affirmation of our proposed method. The summary is as follows:

- Novelity of the paper is good, encourging results. (z95x)
- Intriguing. (iHcA)
- Reasonable solution, acceptable performance. (6Bc6)
- Tackling the cross-platform challenges. (g8nh)

Additionally, we are delighted to receive recognition from the reviewers for the clarity of our writing in the paper. For example:
- Well-written and flows smoothly. (iHcA)
- The structure of this paper is well organized, with clear and concise explanations of the concepts. (g8nh)


It also seems that all the reviewers have provided suggestions for areas that require our further elaboration. In response to feedback, we provide a general response here to emphasize our core contributions, individual responses below to address each reviewer's concerns, and an updated manuscript.


**Motivations and background.**
To our best knowledge, existing high-performance AI **video codecs** [1,2,3,4,5,6,7,8,9,10] are all based on **entropy model**, the core principle of which is to transmit data at a lower bitrate by more accurate estimation of the probability distribution, i.e., the output of the entropy model. To accurately transmit data using arithmetic coding, it is necessary to ensure that the probability distributions used during compression and decompression are identical. However, in practice, the entropy model runs on **different platforms**. At this point, when running the entropy model in floating point operations (either half-precision or single-precision, i.e., FP16 or FP32), there is usually a slight deviation, caused by unavoidable system errors, which leads to decoding errors.

Consequently, some existing **image codecs** quantize the entropy model and then perform integer computation to ensure consistency at the encoding and decoding ends[11,12,13]. Additionally, there is also a **video codec** that partially solves the problem of computational errors in the entropy model under floating point operations (e.g., FP32), which is the first cross-platform video codec [10]. This method corrects the output of the entropy model on different platforms by transmitting an additional bitstream of calibration information.

**Our contributions.**
In this paper, we propose a novel **codebook-based** video codec, as an alternative to the **entropy model-based** video codecs. Since our approach eliminates the dependence on the entropy model, there is inherently no need for the computations at the encoding and decoding ends to be identical. Then our codebook-based video codec is able to run cross-platform. Thus, we have transformed the **impossible** cross-platform task into a **possible** one. Additionally, we propose a WCA-based context model to fuse the context between frames. Due to the absence of autoregressive modeling and optical flow alignment, we can design an extremely minimalist framework that can greatly benefit computational efficiency.

**Limitations.**
Our proposed method constrains the bitrate by manually designing the codebook size. Thus it does result in the performance of our method only surpassing standard codecs such as H.264 and H.265 to some extent for the time being. This still falls short of the state-of-the-art (SOTA) entropy model-based AI video codecs. However, as stated in the conclusion section of this paper, we will carefully investigate in the future to incorporate content-adaptive bitrate constraints into the existing codebook-based framework to further improve the performance.


**Details description.**
We have revised some sections based on the review suggestions, and the updated sections are marked in red font. The revised paper has been uploaded to the system, and screenshots of the corresponding sections have been attached in the respective replies for easy review by the reviewers. Some of the attachments in our responses have also been provided through anonymous links.

---

> ### Author Response · Authors · 2023-11-20
> **For all reviewers (2/2)**
>
> **References**
>
> [1] Eirikur Agustsson, David Minnen, Nick Johnston, Johannes Balle, Sung Jin Hwang, and George Toderici. Scale-space flow for end-to-end optimized video compression. In Proceedings of the IEEE/CVF Conference on Computer Vision and Pattern Recognition, pp. 8503–8512, 2020.
>
> [2] Zhihao Hu, Zhenghao Chen, Dong Xu, Guo Lu, Wanli Ouyang, and Shuhang Gu. Improving deep video compression by resolution-adaptive flow coding. In Computer Vision–ECCV 2020: 16th European Conference, Glasgow, UK, August 23–28, 2020, Proceedings, Part II 16, pp. 193–209. Springer, 2020.
>
> [3] Jiahao Li, Bin Li, and Yan Lu. Deep contextual video compression. Advances in Neural Information Processing Systems, 34:18114–18125, 2021a.
>
> [4] Jiahao Li, Bin Li, and Yan Lu. Hybrid spatial-temporal entropy modelling for neural video compression. In Proceedings of the 30th ACM International Conference on Multimedia, pp. 1503–1511, 2022.
>
> [5] Jiahao Li, Bin Li, and Yan Lu. Neural video compression with diverse contexts. arXiv preprint arXiv:2302.14402, 2023.
>
> [6] Guo Lu, Wanli Ouyang, Dong Xu, Xiaoyun Zhang, Chunlei Cai, and Zhiyong Gao. Dvc: An end-to-end deep video compression framework. In Proceedings of the IEEE/CVF Conference on Computer Vision and Pattern Recognition, pp. 11006–11015, 2019.
>
> [7] Guo Lu, Xiaoyun Zhang, Wanli Ouyang, Li Chen, Zhiyong Gao, and Dong Xu. An end-to-end learning framework for video compression. IEEE transactions on pattern analysis and machine intelligence, 43(10):3292–3308, 2020.
>
> [8] Fabian Mentzer, George Toderici, David Minnen, Sung-Jin Hwang, Sergi Caelles, Mario Lucic, and Eirikur Agustsson. Vct: A video compression transformer. arXiv preprint arXiv:2206.07307, 2022.
>
> [9] Jinxi Xiang, Kuan Tian, and Jun Zhang. Mimt: Masked image modeling transformer for video compression. In The Eleventh International Conference on Learning Representations, 2022.
>
> [10] Kuan Tian, Yonghang Guan, Jinxi Xiang, Jun Zhang, Xiao Han, and Wei Yang. Towards real-time neural video codec for cross-platform application using calibration information. arXiv preprint arXiv:2309.11276, 2023.
>
> [11] Johannes Balle, Nick Johnston, and David Minnen. Integer networks for data compression with latent-variable models. In International Conference on Learning Representations, 2019.
>
> [12] Esin Koyuncu, Timofey Solovyev, Elena Alshina, and Andr ́e Kaup. Device interoperability for learned image compression with weights and activations quantization. In 2022 Picture Coding Symposium (PCS), pp. 151–155. IEEE, 2022.
>
> [13] Dailan He, Ziming Yang, Yuan Chen, Qi Zhang, Hongwei Qin, and Yan Wang. Post-training quantization for cross-platform learned image compression. arXiv preprint arXiv:2202.07513, 2022.

---

### Meta-Review · Area_Chair_39fH · 2023-12-10

**Metareview:**

The authors propose a cross-platform video compression framework based on codebooks. It avoids the need for entropy modelling in probability distribution estimation by leveraging codebook-based approaches and proposes a conditional cross-attention module to obtain the context between frames avoiding the use of optical flow for alignment. The comparisons are against standard codecs.

- Reviewer iHcA is asking for "more RD-performance comparison with existing neural video compression methods", reporting on "decoding efficiency", "more consideration of cross-platform scenario when encoding/decoding", "more comparison with existing calibration methods"
- Reviewer z95x provides a very brief review and asks for adding more recent literature and a detailed analysis of figure 5
- Reviewer 6Bc6 finds "the major architecture and designs lack sufficient novelty", "the experimental results are unconvincing", and that GOP size should be set to 32 and discussed for fairness.
- Reviewer g8nh appreciates that "the experimental results do not effectively validate the method's effectiveness", "authors used GOP=12 for H.264 and H.265 configurations, but the model in the paper used GOP=32"

The authors provides responses to all the reviewers, however aside of Reviewer z95x (rating 8 with a very brief review), the other Reviewers remain in borderline/reject (5,5,6) expressing multiple concerns.

The meta-reviewer, after carefully checking the reviews, the discussions, and the paper, agrees that the paper requires a significant revision as it lacks in aspects pointed out by the reviewers, like novelty, clarity and details, and sufficient experiments.

The authors are invited to benefit from the received feedback and further improve their work.

**Justification For Why Not Higher Score:**

The paper does not meet the acceptance bar due to the multiple weaknesses and issues pointed out by the reviewers. A major revision and another review round is necessary. The most favourable review unfortunately has the least substance from the least confident and experienced reviewer z95x.

**Justification For Why Not Lower Score:**

N/A

---

### Decision · Program_Chairs · 2024-01-16

Reject